# Role of sublimation and riming in the precipitation distribution in the Kananaskis Valley, Alberta, Canada

Émilie Poirier[1], Julie M. Thériault[1], and Maud Leriche[1,2]

[1]Centre ESCER, Departement of Earth and Atmospheric Sciences, Université du Québec à Montréal, Montréal, Québec, Canada
[2]Laboratoire d'Aérologie, CNRS, Université Paul Sabatier, Toulouse, France

**Correspondence:** Julie M. Thériault (theriault.julie@uqam.ca)

**Abstract.**

The phase of precipitation and its distribution at the surface can affect water resources and the regional water cycle of a region. A field project was held in March-April 2015 on the eastern slope of the Canadian Rockies to document precipitation characteristics and associated atmospheric conditions. During the project, 60% of the particles documented were rimed, in relatively warm and dry conditions. Rain-snow transitions also occurred aloft and at the surface in sub-saturated conditions. Ice-phase precipitation falling through a saturated atmospheric layer with temperatures >0°C will start melting. In contrast, if the melting layer is sub-saturated, the ice-phase precipitation undergoes sublimation, which increases the depth of the rain-snow transition. In this context, this study investigates the role of sublimation and riming on precipitation intensity and type reaching the surface in the Kananaskis Valley, Alberta, during March-April 2015. To address this, a set of numerical simulations of an event of mixed precipitation observed at the surface was conducted. This event on 31 March 2015 was documented with a set of devices at the main observation site (Kananaskis Emergency Services, KES) including a precipitation gauge, disdrometer, and micro rain radar. Sensitivity experiments were performed to assess the impacts of temperature changes from sublimation and the role of the production of graupel (riming) aloft on the surface precipitation evolution. A warmer environment associated with no temperature changes from sublimation leads to a peak in the intensity of graupel at the surface. When the formation of graupel is not considered, the maximum snowfall rate occurred at later times. Results suggest that unrimed snow reaching the surface is formed on the western flank and is advected eastward. In contrast, graupel would form aloft in the Kananaskis Valley. The cooling from sublimation and melting by rimed particles increases the vertical shear near KES. Overall, this study illustrated that the presence of graupel influenced the surface evolution of precipitation type in the valley due to the horizontal transport of precipitation particles.

## 1 Introduction

The phase of precipitation can lead to major disasters such as the Calgary 2013 flooding event (Milrad et al., 2015; Liu et al., 2016). In this particular event, the heavy rain generated rainfall runoff at low and mid elevations, but it was supplemented by rain-on-snow runoff at high elevations due to a late lying snowpack (Pomeroy et al., 2016). The rain-on-snow caused by a higher than usual 0°C isotherm was one of the many factors that led to this catastrophic flooding.

The rain-snow boundary, also called the precipitation transition region, is the area characterized by mixed precipitation bounded by only rain and only snow at the surface and aloft, respectively. The top of the boundary corresponds to the top of the melting layer aloft, which is associated with a maximum reflectivity value called the radar bright band (Fabry and Zawadzki, 1995) and the base of the boundary is when all snow/graupel has melted into rain. Marwitz (1983) and Marwitz (1987)studied these rain-snow transitions in mountainous areas using observations over the Sierra Nevada, USA. They observed that the height of the radar bright band, which is associated with the top of the precipitation transition region, decreased by 400-600 m while approaching the mountain barrier, corresponding to a lower $0°C$ isotherm near the barrier. Simulations were also used to study the lowering of the rain-snow boundary on a mountain windward slope (Minder et al., 2011). These simulations identified three two physical mechanisms influencing the location of the rain-snow boundary along the mountainside: cooling by melting of solid ice-phase particles and adiabatic cooling of rising air. The distance associated with complete melting of ice-phase precipitation was also an important factor.

In a saturated environment, diabatic cooling due to melting of ice-phase precipitation falling in a warm layer (T>0°C) can lead to a change in the valley wind flow. This was observed in the Toce river valley in the Italian Alps during the Mesoscale Alpine Program (MAP, Steiner et al., 2003) and during the 2010 Vancouver Olympics in the Whistler area (Thériault et al., 2012, 2015). However, Zängl (2007) used numerical 3D simulations to demonstrate that the cooling by melting of snow was of less importance in creating the down-valley flow for the same event as Steiner et al. (2003) because of the impact of cooling associated with sublimation.

Because ice-phase precipitation melts only when the wet-bulb temperature is >0°C, it can reach the surface at above-freezing temperature in a dry environment. For example, a few studies reported ice-phase precipitation at surface air temperature of 4-6°C (e.g. Matsuo et al., 1981; Harder and Pomeroy, 2013). Few studies addressing the effects of cooling by sublimation in winter storms exist, especially in mountainous regions. For instance, Clough and Franks (1991) examined the evaporative processes in frontal and stratiform precipitation. They showed that sublimation of ice particles was an efficient thermodynamic process. Parker and Thorpe (1995) studied the role of snow sublimation on frontogenesis and showed that the cross-frontal flows in the vicinity of the sublimation were strongly modified, and a mesoscale downdraft was produced below the synoptic frontal surface. Barth and Parsons (1996) highlighted that sublimation of snow and rimed particles played an important role in the modelled evolution of a narrow cold-frontal rain band.

Few studies have examined precipitation features in northern Canada, in relatively dry areas. Burford and Stewart (1998) suggested that sublimation was the main process responsible of relatively low precipitation amounts observed at Inuvik and Tuktoyaktuk (Northwest Territories). Furthermore, Stewart et al. (2004) examined precipitation events at Fort Simpson (Northwest Territories) and found that hydrometeors were mainly single crystals and aggregates. The absence of rimed particles could explain the low precipitation amounts as single crystals and aggregates are more likely to sublimate while falling to the surface. In contrast, Henson et al. (2011) and Fargey et al. (2014) characterized precipitation types over Baffin Island, Nunavut, showing that rimed particles, aggregates, and snow pellets were very common even during light precipitation events. They suggested that the development of rimed and large particles increased their likelihood of reaching the surface through the drier sub-cloud layer.

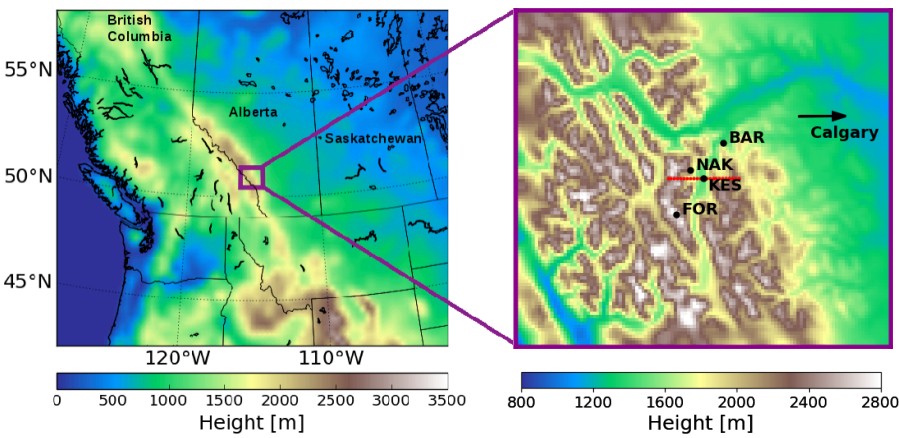

**Figure 1.** Area of interest (left) and 1-km mesh domain (right) used for the numerical simulations with the WRF model. BAR stands for the Barrier Lake research station, NAK for Nakiska ski area, KES for the Kananaskis Emergency Services site and FOR for Fortress Mountain. Red line on the right panel indicates the position of the cross section used in Fig. 6 and 8.

In this context, to better understand the processes leading to surface precipitation on the lee side of the Canadian Rockies, a field campaign was held during March-April 2015 in the Kananaskis Valley, Alberta (Fig. 1). The goal of this field campaign was to document precipitation and associated weather conditions in that region (Thériault et al., 2018). Given the importance of precipitation phase in this area, there is a need to improve our understanding of the physical processes leading to rain-snow

transition in these particular sub-saturated areas. The goal of this study is to investigate the role of sublimation and riming on the precipitation intensity and types reaching the surface in the Kananaskis Valley, Alberta. To address this, numerical simulations using the Weather Research and Forecasting (WRF) model (Skamarock and Klemp, 2008) were conducted. A well-documented case study associated with mixed precipitation reaching the surface in the Kananaskis valley was chosen from the 2015 field campaign mentioned above. After verifying that the model is able to represent this observed case study,

numerical simulations are used to investigate physical processes producing the distribution of precipitation in the Kananaskis area.

This paper is structured as follows. Section 2 provides an overview of the field project and describes the case study used in this paper. The methodology, including the model configuration, the sensitivity experiments, and the data analysis, is explained in Section 3. Results from the control simulation, the role of sublimation, and the formation of rimed snow are summarized in

Sections 4, 5 and 6, respectively. Finally, a summary and conclusion are given in Section 7.

## 2   Overview of the case study

During the Alberta Field Project held in the Kananaskis Valley in March-April 2015, a total of 17 precipitation events were documented (Thériault et al., 2018). These were associated with rain or snow only, as well as a mixture of precipitation. Ice-

phase precipitation was reported at the surface at temperatures up to 9°C but in very dry conditions (∼45% relative humidity), also noted by Harder and Pomeroy (2013), and most of them were rimed (∼60%).

Most of the observations were collected at the Kananaskis Emergency Services (KES) site located a few kilometers southeast of the Nakiska ski area (NAK) and about 15 km south of the Barrier Lake research station (BAR) (Fig. 1), and are now detailed. To characterize the atmospheric conditions (temperature and relative humidity) aloft, a sounding system was used, and balloons were launched at every 3 h during precipitation events. The precipitation layer aloft was characterized using a Micro Rain Radar (MRR2, Klugmann et al., 1996). MMR2 gives the temporal evolution of the vertical profile of the reflectivity and Doppler velocity; note that this measurement is affected by the signal attenuation due to e.g. the bright band (Matrosov et al., 2008). A GEONOR weighing precipitation gauge (Rasmussen et al., 2011) was used to measure the liquid equivalent amount of precipitation. An optical disdrometer, OTT Parsivel 2 (Battaglia et al., 2010), was used characterized the type of hydrometeor by measuring the fall-speed and diameter of precipitation particles. The precipitation types were automatically diagnosed using the Ishizaka et al. (2013) method and the optical disdrometer data (Thériault et al., 2018). Manual observations of weather conditions including precipitation types were also reported in a systematic manner. The manual method is more precise because one can estimate the degree of riming and the exact crystal types. The Ishizaka et al. (2013) method gives a good idea of the degree of riming but it is not possible to diagnose the type of ice crystal because of the bin sizes. Basic meteorological measurements were also available (pressure, wind speed and direction, temperature, dew point temperature). Finally, vertical profiles of basic meteorological features were obtained using a Kestrel attached to a ski pole and a GPS (Thériault et al., 2014) to characterize rain-snow transitions at NAK and at Fortress Mountain (FOR). Further details about the field campaign are given in Thériault et al. (2018).

The well-documented weather event that occurred on 31 March 2015 was chosen for this study for three main reasons. First, all of the weather instruments deployed at KES were operational. Second, a mixture of precipitation types and phase transition in sub-saturated conditions occurred at the surface so it is possible to investigate the role of melting and sublimation of ice hydrometeors. Finally, detailed measurements on the height and width of the transition have been conducted along FOR using the 'car-sonde' technique described in Thériault et al. (2018).

On 31 March 2015, a weather event associated with a rain-snow boundary along the mountainside occurred in the Kananaskis Valley. The sounding launched at 2100 UTC shows sub-saturated conditions near the surface at the KES site (Fig. 2). The MMR2 reflectivity profiles (Fig. 3a) show precipitation reaching the surface for about 2 hours. The radar reflectivity bright band (>30 dBZ) is located at the elevation where ice-phase precipitation started to melt (Fig. 3a). Measurements along FOR using the 'car-sonde' technique indicated that the top of the rain-snow boundary was at 1750 m ASL. The rain-snow transition is located about 200 m below the 0°C isotherm, which confirms that ice-phase precipitation was not melting until the level associated with a wet-bulb temperature, $T_w$, >0°C was reached (Harder and Pomeroy, 2013).

The surface temperature at 2100 UTC 31 March 2015 decreases from 12°C to 3°C whereas the relative humidity increases up to the onset of precipitation from around 25% to around 75% (Fig. 3c). Wind speed is generally weak throughout the precipitation event, with variable directions (Fig. 3d). Manual observations at the KES site show that light rain started at 2030

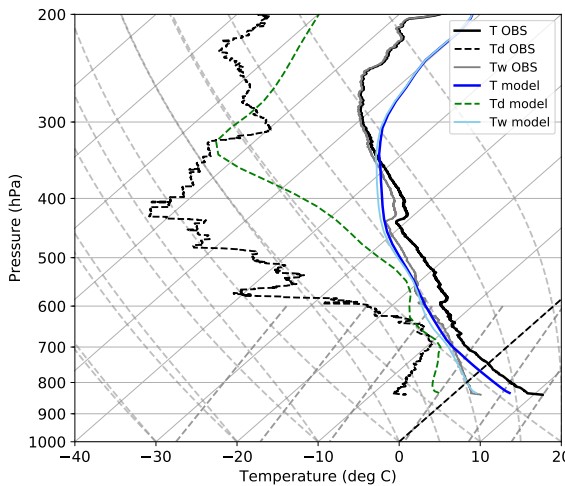

**Figure 2.** Vertical profiles of air temperature (T, solid line), dew point temperature (Td, dashed line) and wet-bulb temperature (Tw, light colour) at 2100 UTC 31 March 2015 at the KES site. The measurement (OBS) and the control simulation (model) described in section 3.1. are represented by blue and black/grey lines, respectively.

UTC 31 March 2015, changing to a mixture of rain, snow and graupel between about 2150 UTC and 2215 UTC, then to a brief period of only rain (Fig. 3f). The automatic diagnostic of precipitation types supports this.

## 3 Description of the simulations

### 3.1 Model configuration

Three-dimensional (3D) simulations are performed using the WRF model, version 3.7.1 (Skamarock and Klemp, 2008), with initial and boundary conditions provided by the North American Regional Reanalysis (NARR) data from the National Center for Environmental Prediction (NCEP) (Mesinger et al., 2006). The boundary conditions forcing is done every 3 hours. Two-way nesting with four nested grids (27 km, 9 km, 3 km and 1 km) is used to perform high-resolution simulations over the Kananaskis valley. The high-resolution domain is shown in Fig. 1; it has 118 x 106 grid points. The following analysis of simulated results

will focus only on outputs from the high-resolution domain. The control run (CTR hereafter) and the sensitivity tests are conducted with the two-moment bulk microphysics scheme of Milbrandt and Yau (2005a, b) to predict cloud and precipitation.

Other parameterizations used in the simulations include the Rapid Radiative Transfer Model (RRTMG) with the Monte Carlo Independent Column Approximation (MCICA) method of random cloud overlap scheme (Iacono et al., 2008) for longwave and shortwave radiation. Also, the Noah Land Surface Model (Tewari et al., 2004) with soil temperature and moisture in four

layers, fractional snow cover and frozen soil physics is used. The planetary boundary layer is parameterized in the simulations

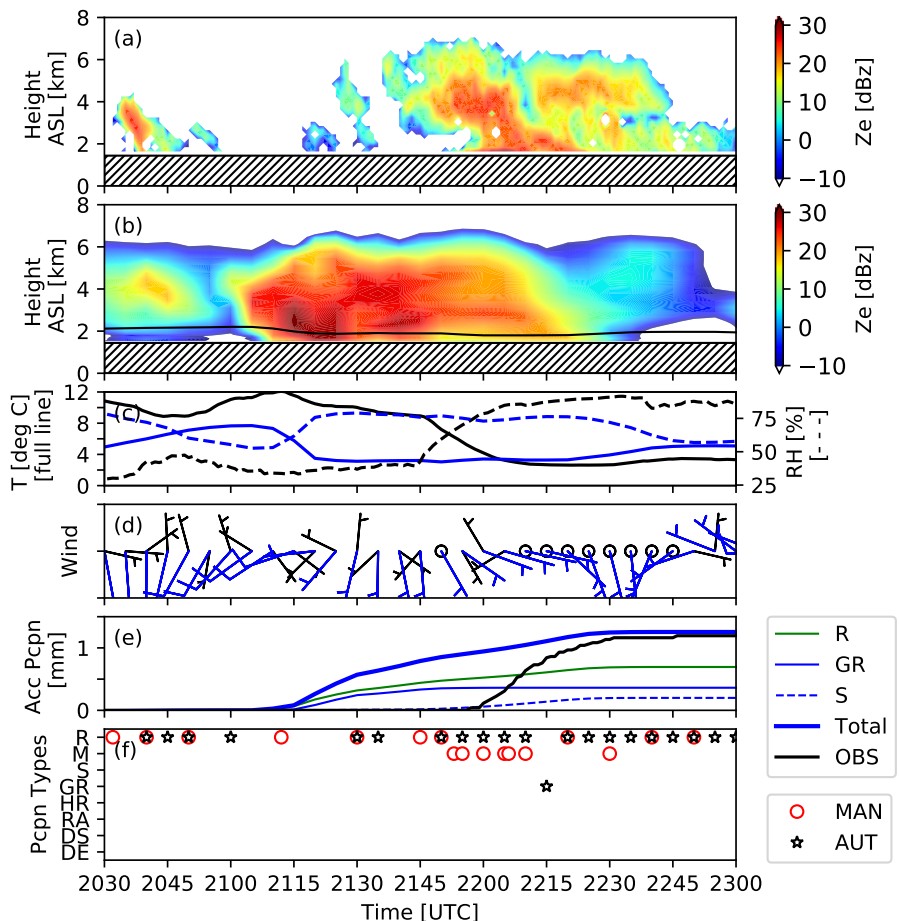

**Figure 3.** Atmospheric conditions and precipitation fields during the 31 March 2015 event at KES. (a) Reflectivity field measured by the Micro Rain Radar (MRR2) and (b) estimated by the model (CTR). Reflectivity values >30 dBZ are associated with the radar reflectivity bright-band; (c) surface temperature (T) and relative humidity (RH) observed (black line) and simulated (blue line); (d) wind speed and direction using wind barbs, where the observed is black and simulated is blue (an empty circle is wind speed rounded at 0 knots, a short bar is rounded at 5 knots); (e) unadjusted liquid equivalent accumulated precipitation observed (black line, OBS) and simulated (bold blue line for total, green line for rain, thin blue line for graupel and dashed blue line for snow), and (f) the type of precipitation observed manual (MAN) and automatically (AUT) at KES. These are rain (R), graupel (GR), snow (S), mixed precipitation (M), heavily rimed snow (HR), rimed aggregates (RA), dry snow (DS) and dendrites (DE). Simulated results are for the CTL run. Adapted from Thériault et al. (2018).

with the Yonsei University scheme, which uses the non-local K approach with an explicit entrainment layer and a parabolic K profile in the unstable mixed layer, where K is the vertical diffusion coefficient (Hong et al., 2006). Cumulus parameterization is used on the coarser grid only (27 km) with the Kain-Fritsch scheme (Kain, 2004).

To have the maximum number of vertical levels within the melting layer, 56 vertical levels are used where the grid spacing varies from 50 to 320 m in the first 2 km and is between 320 and 340 m at higher levels. The simulation on the coarser grids (27, 9 and 3 km) starts at 1500 UTC 31 March 2015, 3 hours prior to the higher resolution grid (1 km), which starts at 1800 UTC 31 March 2015. The simulations are integrated for a total of 12 h and 9 h, respectively. The time step used is 90 s on the coarser grid (27 km) decreasing with a ratio of 3 between each nested grid to 3.33 s on the higher resolution grid (1 km).

## 3.2 Description of the microphysics scheme and modifications

The two-moment microphysics scheme predicts the mass mixing ratio and the total number concentration of inverse exponential size distribution of six hydrometeor categories: cloud droplets, rain, ice crystals, snow, graupel, and hail. Each category is described by an assumed mass-diameter relationship and an associated fall speed. The evolution of clouds and precipitation is based on many microphysical processes that are mainly divided into cold and warm processes in the microphysics scheme. In this study we focus on the sublimation and melting of ice, snow and graupel, as well as the impact of the presence of graupel. This last process includes the collision/coalescence of ice crystals and snow with cloud droplets or raindrops leading to rimed particles. This parameterization differs among bulk microphysics schemes. For example, Milbrandt and Yau (2005a, b) follow Murakami (1990) to parameterize the snow-graupel conversion. It is based on the rate of collection of snow/ice with cloud droplets as well as vapor deposition. The change from the snow category to graupel category involves a sharp increase in density (100 to 400 kg/m$^3$) and, in turn, an increase in the fall velocity ($\sim$ 1 to 3 m s$^{-1}$). Hence, the mass of snow can increase aloft without falling faster until it is converted into graupel. Pre-defined hydrometeor categories are a limitation of bulk microphysics schemes. A more detailed description of the conversion process as well as all processes are given in Milbrandt and Yau (2005a, b).

Since the area of interest for this study is located in a sub-saturated environment, the scheme is modified to allow snow to sublimate at all temperatures. In the original Milbrandt and Yau (2005a, b) two-moment scheme, snow sublimation can only occur when the temperature is <0°C, while graupel can sublimate at all temperatures. Some modifications were made to this original scheme (Poirier, 2017). Given that graupel can sublimate at temperatures >0°C, the same equation was used for snow, which is:

$$QVD_{vs} = \frac{1}{AB_i} \left[ 2\pi(S_i - 1)N_{0s}VENT_s - \frac{L_sL_f}{K_aR_vT^2}QCL_{cs} \right] \tag{1}$$

where

$$AB_i = \frac{L_s^2}{K_aR_vT^2} + \frac{1}{\rho q_{is}\psi} \tag{2}$$

is the thermodynamic function. Also, $S_i$ is the saturation ratio with respect to ice, $N_{0s}$ is the intercept parameter for snow, $VENT_s$ is the mass-weighted ventilation factor (Ferrier, 1994), Ka is the thermal conductivity of air, $R_v$ is the gas constant for water vapour, $T$ is the temperature of air, $\rho$ is the density of air, $q_{is}$ is the saturation vapour mixing ratio with respect to ice and $\psi$ is the diffusivity of water vapour in air.

The sublimation rate equation was moved in the microphysics scheme so that snow and graupel sublimation are computed in the same conditions, at all air temperatures. The function polysvp was also corrected in the microphysics scheme to calculate the saturation vapour pressure properly at all temperatures. It has been fixed in the following version of WRF.

## 3.3 Description of the sensitivity experiments

The control simulation (CTR) is conducted using the modified microphysics and model configuration described in section 3.1 and 3.2. To estimate the impact of temperature changes on the amount and types of precipitation at the surface, three sensitivity experiments were performed while neglecting the diabatic heating/cooling due to the precipitation phase transition and no graupel formation. The temperature tendency equation is:

$$\frac{dT}{dt} = \frac{1}{\Delta t}\left\{ \frac{L_f}{c_{pd}}\left( \begin{array}{c} \Delta QCLcs + \Delta QCLcg + \Delta QCLch \\ + \Delta QCLri + \Delta QCLrs + \Delta QCLrg \\ + \Delta QCLrh + \Delta QFZci + \Delta QFZrh \\ - \Delta QMLir - \Delta QMLsr \\ - \Delta QMLgr - \Delta QMLhr \end{array} \right) + \frac{L_s}{c_{pd}}\left( \begin{array}{c} \Delta QNUvi + \Delta QVDvi + \Delta QVDvs \\ + \Delta QVDvg + \Delta QVDvh \end{array} \right) \right\} \quad (3)$$

where $L_f$ is the latent heat of fusion, $L_s$ is the latent heat of sublimation, $c_{pd}$ is the specific heat of dry air and $Q$ is for mixing ratio. The types of mixing ratios are noted by $CL$ for collection, $FZ$ for freezing, $ML$ for melting, $NU$ for nucleation, $VD$ for diffusional growth (positive) or sublimation (negative) and the subscripts $(c, r, i, s, g, h, v)$ represent cloud droplets, rain, ice, snow, graupel, hail and water vapor.

    The three key sensitivity experiments are:

1. NO_MLT: The diabatic cooling of melting snow (QMLsr) and graupel (QMSLgr) were set to zero in the temperature tendency equation (Eq 1). Hence, snow and graupel were allowed to melt into rain but no energy was extracted from the environment to melt the particles.

   2. NO_SBL: The diabatic cooling of sublimation of snow (QVDvs) and graupel (QVDvg) was set to zero in the temperature tendency equation (Eq 1).

3. NO_GRPL: Since graupel was often reaching the surface at KES during the Alberta field project (Thériault et al., 2018), another simulation was performed. The initiation of graupel was suppressed by turning the production of graupel off (grpl_ON= false). It was also ensured that there were no sources or sinks, hence, no warming from the cloud droplets freezing on the solid particles (snow or/and ice) and no sublimation of graupel since none was produced.

### 3.4 Data analysis

The data are analyzed in a systematic manner. First, the CTR simulation is compared to available observations such as wind speed and direction, temperature, relative humidity, height of the rain-snow transition, as well as precipitation amount and types collected during the field project. The time evolution of mass content of ice crystals, cloud, rain, graupel and snow are analyzed at the grid point closest to the KES site. To analyse precipitation aloft at KES and across the Kananaskis Valley, a vertical cross section is plotted and the mass content of hydrometeors, as well as the vertical air motion, are investigated. Second, the CTR simulation is compared to the three sensitivity experiments: the simulation without the temperature change from melting of snow and graupel (NO_MLT), the simulation without the temperature change from sublimation (NO_SBL), and the simulation without graupel (NO_GRPL). Finally, the impact of wind direction and precipitation types formed aloft on the precipitation amounts and types reaching the surface is investigated.

## 4 Analysis of the control run

### 4.1 Comparison with observations

The CTR simulation is compared to observations to verify that atmospheric conditions are sufficiently well represented to use the simulations as a qualitative analysis tool of physical processes. The simulated liquid equivalent accumulated precipitation is compared to observations in Fig. 4. Comparison shows good agreement at KES and NAK but an overestimation by the model near FOR (Fig. 4d). The gradient of precipitation along the mountainside is well represented, showing that rain accumulated in the valley (Fig. 4c). Higher amounts of graupel (Fig. 4b) are produced at higher elevations where the conditions for riming are more favorable because of the presence of cloud droplets (Milbrandt and Yau, 2005a, b) . Both observations and CTR simulation show that the precipitation amount accumulated at KES is relatively low during this event and dominated by rain (Fig. 4.c) with snow at high elevations (Fig. 4a).

Concerning the general meteorological parameters, the CTR run shows similar patterns as the observations at KES (cf. Fig. 2 and 3). The vertical structure of the temperature and dewpoint are similar but the model is mainly colder and moister than the observations. The wet-bulb temperature is, however, similar (Fig. 2). Although, the timing of the precipitation differs. The simulated and observed relative humidity are similar, and even if temperatures are different before the onset of precipitation, they reach similar values during the precipitation event (cf. Fig. 3c). The wind direction is highly variable, but both the simulation and observation have southerly components before the onset of precipitation while the simulations exhibit slightly stronger winds during the event (cf. Fig. 3d). Ice-phase precipitation is simulated at temperatures >0°C in the Kananaskis area as reported during the field project. Precipitation amounts simulated at KES are very low and reach 1.3 mm during the simulated event in agreement with observations (Fig. 3e).

The rain-snow boundary occurred at warmer temperatures than if the environmental conditions were saturated, and is reproduced between 2100 and 2230 UTC 31 March 2015, as measured by the 'car-sonde' at FOR (Thériault et al., 2018), which varied from 1750 m and 1830 m. The simulated height of the melting layer at about 1600 m (Fig. 5e) corresponds to that

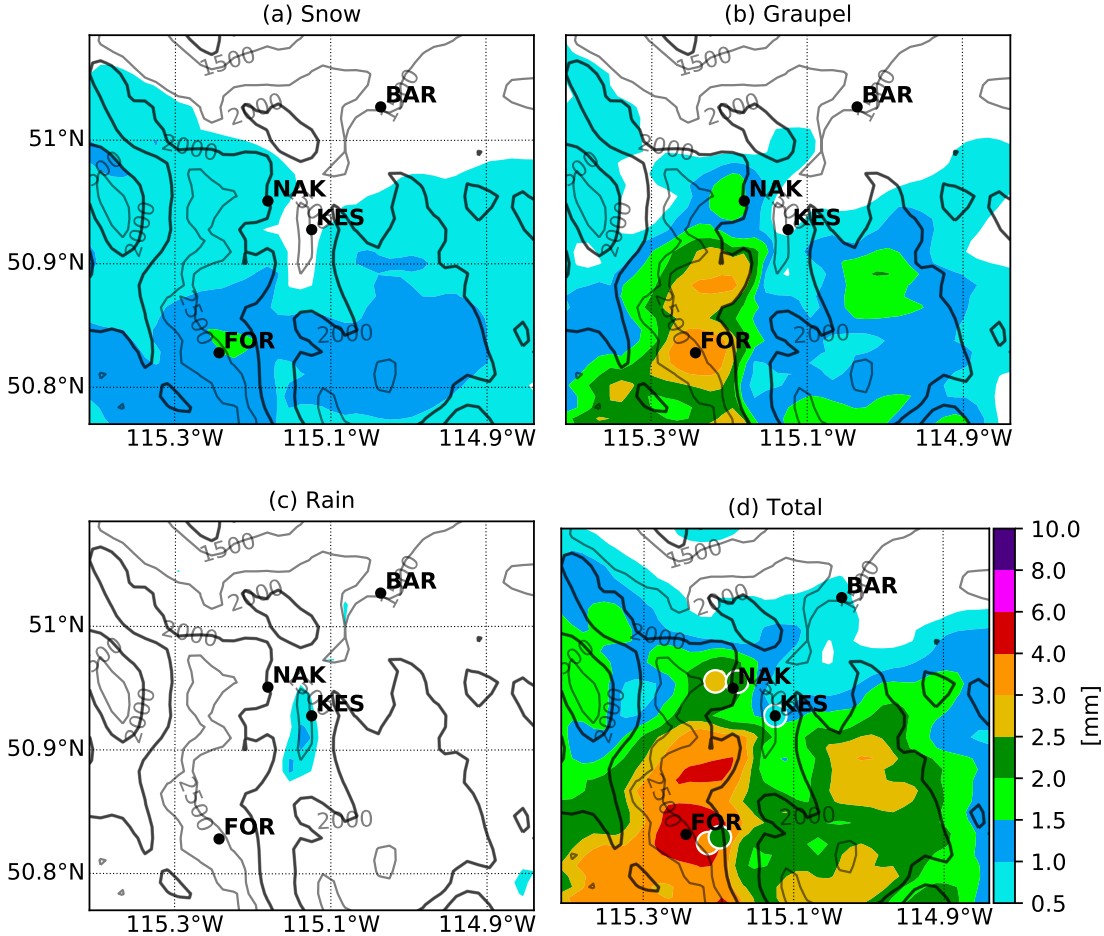

**Figure 4.** Simulated (a) unadjusted accumulated solid precipitation (mm) including (a) snow and (b) graupel, (c) rain and (d) total accumulated precipitation between 2000 UTC 31 March 2015 and 0000 UTC 1 April 2015. The coloured circles in (d) are the observations at 4 locations. These are KES (2.7 mm), Nakiska (2.2 mm), Fortress (3 mm) and Barrier Lake Station (0.8 mm). Accumulated precipitation is in liquid equivalent. The black lines are the topography in meters.

measured by the MRR2 bright band (Fig. 3a). The reflectivity computed is higher than observations and it is difficult to discern the bright band near the surface because of the high reflectivity fields probably produced by graupel (Fig. 3b). The comparison of Fig. 3e and f shows an agreement between the type of hydrometeors simulated and observed, with the predominance of rain and the presence of graupel. We notice that precipitation begins earlier in the simulation than in the observation (almost 1 hour) as shown in Fig. 3e and by the time lag between Fig. 3a and 3b.

In summary, the meteorology at KES is generally qualitatively well simulated during the precipitation event. This statement allows us to use the model to investigate the impact of microphysical processes on the phase and distribution of precipitation at the surface.

## 4.2 Vertical distribution of hydrometeors

The vertical structure of hydrometeors at KES is now investigated. The riming of ice crystals with cloud droplets at 6 km ASL is a minor source of graupel (Fig. 5a). The main source of graupel seems to be riming of snow with cloud droplets based on the order of magnitude mixing ratio of both snow and graupel (Fig. 5e, i and m). Snow occurs aloft throughout the event, but only reaches the surface from 2145 UTC until 2230 UTC at a precipitation rate of $\sim$0.5 mm/h as it sublimates before reaching the surface (Fig. 5q). Rain occurs simultaneously with graupel and snow at the surface throughout the event (Fig. 5q). It corresponds mainly to the type of precipitation reported in Fig. 3f.

The vertical cross-section of hydrometeors when snow starts to reach the surface (2145 UTC) is shown in Fig. 6 across the Kananaskis valley. The maximum mass-contents of ice crystals and cloud droplets aloft occurred on the windward side of mountain slopes (Fig. 6a and e) and are transported across the valley. The location of the maximum amount of ice crystals corresponds to the elevation where snow is formed. At that level, ice crystals interact with cloud droplets to produce graupel aloft (Fig. 6i). Snow is transported eastward by the wind and sublimates (Fig. 6m). The upward air motion leads to the formation of ice crystals (Fig. 6a) and cloud droplets (Fig. 6e) aloft above the westward barrier, which are converted into snow and graupel, and transported downstream (Fig. 6i and m). Clouds (both ice and liquid) and precipitation produced on the westward barrier (-10 km) are transported east of KES. Note that no hail was produced in the simulations.

The remaining analysis will focus on the microphysical processes near KES. In particular, the impact of sublimation and the occurrence of graupel on the formation and evolution of precipitation types and amounts, as well as the wind field are investigated.

## 5 Hydrometeor evolution during the event: CTR versus sensitivity experiments

The roles of phase changes and of the production of graupel on precipitation amounts and types reaching the surface at KES are investigated by comparing the CTR simulation with sensitivity experiments (NO_MLT; NO_SBL; NO_GRPL). First, for NO_MLT, differences with CTR on the surface precipitation intensity and type at KES are minor (Fig. 6, first and second column). For example, NO_MLT simulation produces slightly less precipitation than CTR, but the precipitation types and their evolution is similar (Fig 5q and r). In contrast, the evolution of precipitation intensity and types varies significantly at KES,

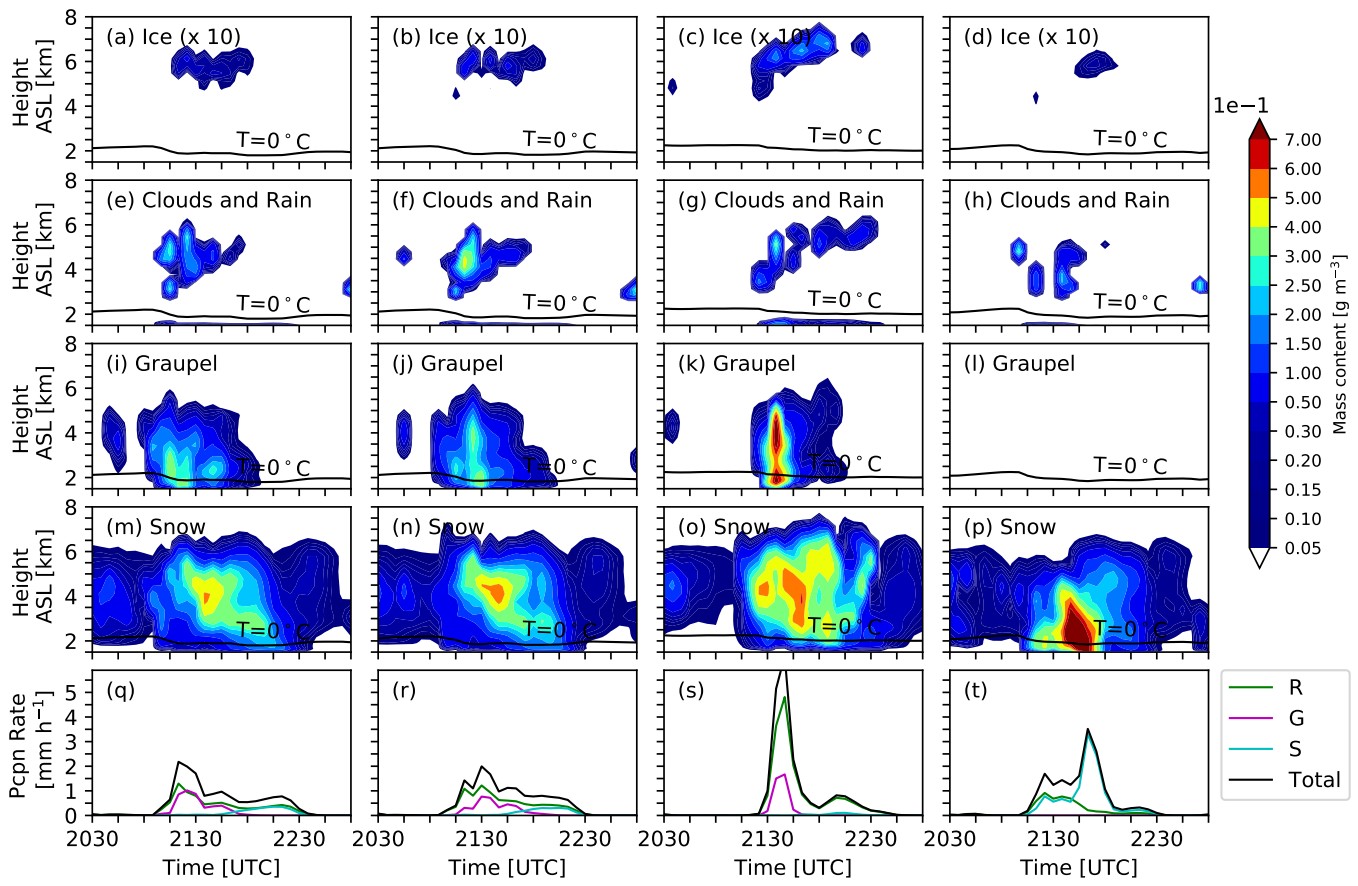

**Figure 5.** Comparison of the time evolution of hydrometeors mass content (g/m$^3$) at the surface and aloft at KES during the 4 simulations conducted for CTR, NO_MLT, NO_SBL and NO_GRPL from left to right. (a-d) is ice mass content (x10 g/m$^3$), (e-h) is clouds and rain mass content, where rain is only formed through melting of ice, so it is only present near the surface, (i-l) is graupel mass content, (m-p) is snow mass content and (q-t) is the surface precipitation rate of rain (R), graupel (G) and snow (S). The 0°C isotherm is indicated by the solid black line on (a-p). Panels a-p have the same colour scale.

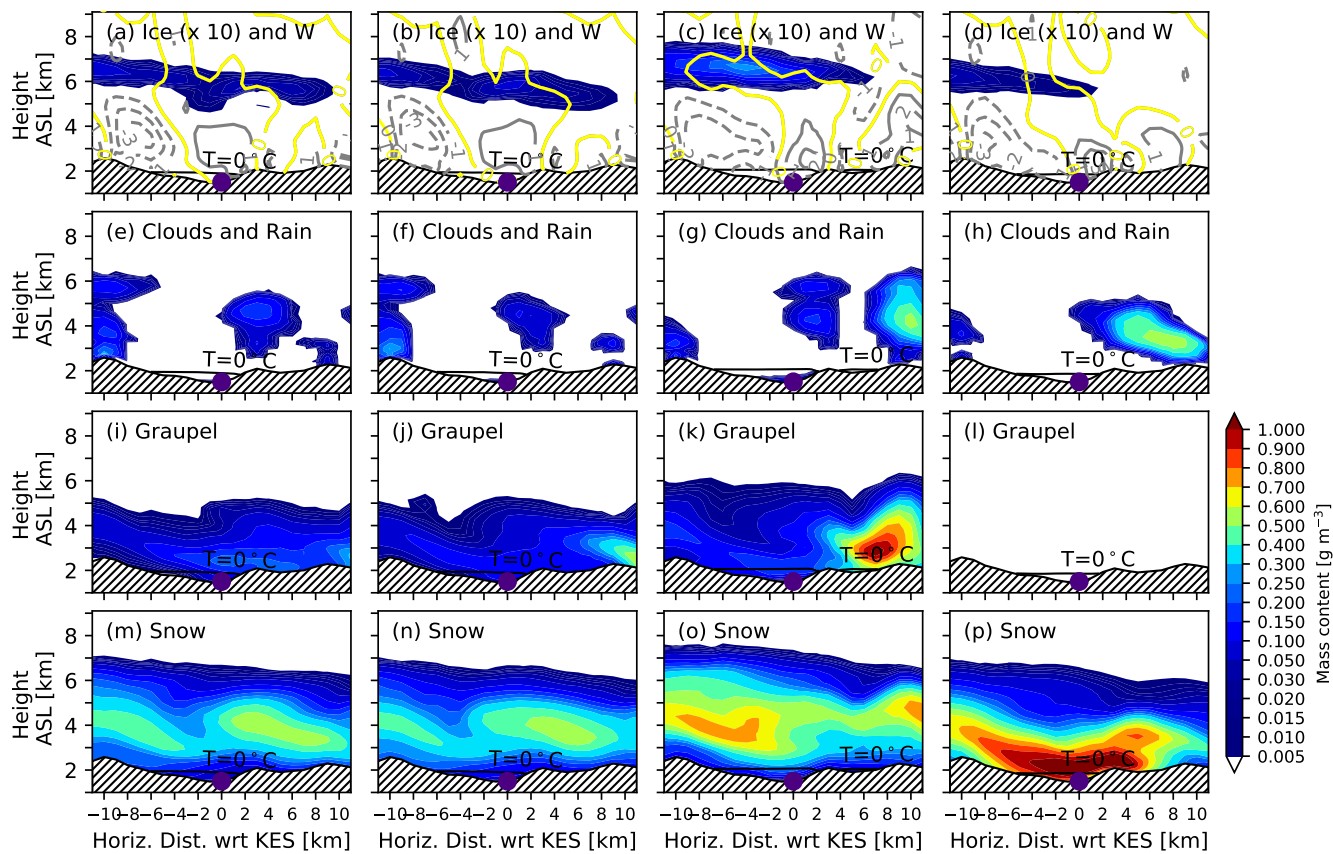

**Figure 6.** Comparison of the vertical cross-section across the Kananaskis Valley along the red line in Fig. 1 showing the mass content of hydrometeors (g/m$^3$ during the 4 simulations conducted for CTR, NO_MLT, NO_SBL and NO_GRPL from left to right. (a-d) is ice mass content (x10 g/m$^3$) with vertical velocity (m/s). The yellow line is 0 m/s, the dashed lines are negative values and solid lines are positive values, (e-h) is clouds and rain mass content, (i-l) is graupel mass content and (m-p) is snow mass content. The 0$^\circ$C isotherm is indicated by the solid black line. Panels a-p have the same colour scale. The location of KES is indicated by the purple dot.

in comparison with CTR for NO_SBL and NO_GRPL (Fig. 5s and t). The peak in precipitation occurred at the beginning of the event (∼2135 UTC) in the warmer environment (NO_SBL) and later during the event (∼2150 UTC) when no graupel were produced (NO_GRPL). Given these findings, the effects of temperature changes from sublimation and riming on the production and the evolution of precipitation are further investigated.

The time series of the vertical evolution of clouds and precipitation for the CTR and the sensitivity experiments at KES are shown in Fig. 5. The distribution of hydrometeors at KES for NO_MLT (Fig. 5 second column) is similar to the CTR with very little change in precipitation and cloud distribution. When no temperature change from sublimation is considered (NO_SBL), the timing of precipitation is delayed in comparison to CTR (Fig. 5 third column). In that case, the ice and liquid water clouds (Fig. 5c and g) persist for a longer time period than CTR. Moreover, the top of the ice cloud extends up to 7 km leading to

graupel formation (Fig. 5k) at higher elevations compared to CTR (Fig. 5). In NO_SBL, the elevation of the 0°C isotherm is higher than CTR because the environmental air is generally warmer (Fig. 5 third column). It produces favourable conditions for ice-phase precipitation to melt into rain before reaching the surface. These statements suggest a link among the maximum precipitation rate at the surface produced in warmer conditions (NO_SBL, Fig. 5 third column) as well as the highest ice crystal mass content aloft (Fig. 5c). For NO_GRPL (Fig. 5 fourth column), ice crystals, cloud droplet and precipitation distribution

aloft, as well as at the surface, differs from their counterparts in the CTR simulation (Fig. 5 fourth column). Less ice crystals and cloud water mass contents are produced aloft compared to CTR. This could be explained by the lack of warming from accretion resulting in colder temperatures, which leads to less water vapour depositional growth for ice crystals and cloud droplets, and less ice nucleation aloft (e.g. Meyers et al., 1992). Below the ice cloud, less cloud droplets are produced for a similar reason. Once graupel are formed, the environmental temperature increases due to the latent heat of fusion from the

freezing of cloud droplets. As graupel fall through sub-saturated conditions, they cool the environment because of sublimation, which alters the distribution of hydrometeors aloft (Fig 5). Finally, in NO_GRPL , the peak in surface precipitation rate (Fig. 5t) is delayed because only relatively slow-falling ice particles such as snowflakes are formed.

     The vertical evolution of hydrometeors when snow starts to reach the surface in CTR (2145 UTC) across the Kananaskis Valley differs for each simulation (Fig. 6). First, for NO_MLT (Fig. 6 second column), no significant difference is observed

with CTR simulation for the reason discussed earlier in this section . Second, for NO_SBL (Fig. 6 third column), more rain reaches the surface in the valley because the environmental temperature is higher in comparison to CTR. For NO_SBL, vertical air motions are stronger on the slope east of KES to produce a deep liquid water cloud (Fig. 6c). The ice cloud is also higher and deeper in NO_SBL due to warmer conditions than in CTR (Fig. 6c). The formation of graupel also affects the distribution of cloud and precipitation in the Kananaskis Valley (Fig 6 fourth column). In NO_GRPL, the ice cloud extends up to KES but

it does not interact with the liquid water cloud, which is formed at lower levels compared to CTR. At this time, the snowfall rate at the surface increases rapidly to reach 3 mm/h and the rain rate decreases. From Fig. 5, the time series of hydrometeor evolution is completely different for NO_GRPL compared to other three cases. The cross-section at 2145 UTC (Fig. 6) is not representative of the precipitation onset but it corresponds to a time when the many types of hydrometeors are simulated aloft.

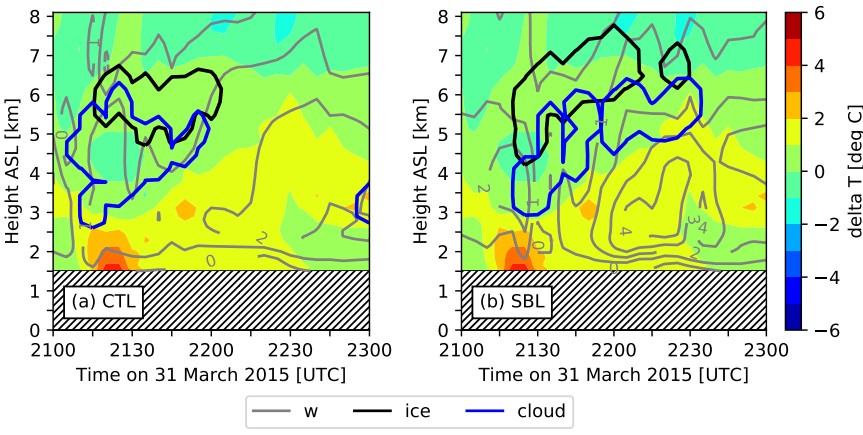

**Figure 7.** Time series of the vertical air motion ($w$, grey dashed lines) as well as the contour delimiting water and ice clouds using a minimum threshold of the mass-content ($5\text{x}10^{-3}$ g/m$^3$ and $5\text{x}10^{-4}$ g/m$^3$, respectively) (blue, respectively, black line) are superimposed for (a) CTR and (b) NO_SBL. The temperature difference between the NO_SBL and the CTR simulations (delta T) is added to each panel.

## 6 Role of sublimation and snow pellet formation

This section will assess the role of sublimation and graupel formation on the vertical and horizontal evolution of precipitation intensity and types in the Kananaskis Valley.

First, neglecting the cooling due to sublimation results in higher temperatures at both the surface and aloft (Fig. 7). This
higher temperature aloft in the NO_SBL run would increase the amount of snow aloft and at the surface but only aloft for graupel. As KES is located on the windward side of the Kananaskis Valley, there is generally upward motion at that location. The warmer conditions in NO_SBL produce more instability and, in turn, stronger upward motion. This upward motion leads to thicker and higher ice clouds and liquid water clouds (Fig. 7b). Comparing CTR to NO_SBL, (Fig. 5q and s) shows that the maximum precipitation occurs at the beginning of the event for CTR. Warmer conditions in NO_SBL delays the onset of
precipitation because of sub-saturated conditions aloft. Once the clouds are formed, precipitation reaches the surface at higher rates at KES because less is being transported eastward. The higher rain rate is due to a higher melting level aloft allowing for complete melting of ice-phase precipitation before reaching the ground.

Second, graupel formation impacts the surface precipitation intensity and types, in particular, by indirectly influencing the wind flow in the valley. For the CTR case, the evolution of the horizontal wind speed between the onset of the precipitation and
the end of the precipitation event (Fig, 8a and d) shows that the direction and magnitude of the wind speed change at the end of the precipitation event in the valley close to KES on the western slope. For the NO_GRPL, this wind shear on the windward side of the mountain is suppressed (Fig. 8c and f) whereas it is maintained in NO_SBL (Fig. 8b and e) and NO_MLT (not shown) with a smaller magnitude in both cases. This suggests that the cooling from sublimation and/or melting of graupel produces denser air that moves down the mountainside. In CTR and NO_GRPL, snow is produced mainly over the western

barrier with respect to KES as shown on Figs. 8a and c at the onset of precipitation. The snow mass-content suggests that snow is transported downwind between the onset and end of the precipitation event (Fig. 8). The change in the zonal wind speed (Fig. 8a,c,d and f) prevents snow from falling at KES in CTR (Fig. 5q). As snowflakes fall at low speed (about 1 m/s), their trajectories are strongly dependent on the prevailing horizontal wind field. Since easterly winds were up to 2 m/s at 2200 UTC in the CTR, very little snow reaches the surface at KES (Fig. 5q). In NO_GRPL, snow reaches the surface (Fig. 5t) because the downslope flow is weaker than in CTR. Note that in the warmer environment (NO_SBL), the flow reversal is weaker than in the colder one (CTR) but stronger than without snow pellet formation (NO_GRPL). Hence, in the warmer environment (NO_SBL), the deviation of the snow-mass content is not as pronounced as in the colder environment. Furthermore, snowflakes are falling much more slowly than graupel (up to 4 times) and will tend to more closely follow streamlines as compared to graupel. This is a possible explanation for the difference in the surface precipitation intensity and types at KES (Fig. 5q-t) and across the Kananaskis Valley (Fig. 8g-i). Moreover, in NO_GRPL, less cloud droplets are produced (Fig. 5) over KES. This is probably due to the lack of warming feedbacks from the production of graupel that is considered in CTR. Due to less sublimation occurring above KES in NO_GRPL, the change in the valley flow field is not as strong as in CTR. This leads to more orographic forcing in NO_GRPL, producing the clouds aloft. In this case, the amount of snow produced above KES is negligible because snow produced aloft is advected downwind. Therefore, snow reaching KES is mainly formed on the western barrier with respect to KES. The precipitation is transported downwind to KES. This explains why the peak in precipitation rate occurs at later times near the end of the event for NO_GRPL (Fig. 5t).

Finally, given that the trajectories of ice-phase precipitation differ among NO_SBL, CTR and NO_GRPL, the latent heating profiles also differ. The trajectories of precipitation particles can explain why more cooling from sublimation occurs in NO_GRPL than CTR. Snow would come from the western barrier and the sublimation would occur along the trajectories while simulations with faster falling particles, would lead to sublimation aloft in the Kananaskis Valley, closer to KES. Note that no lateral shift of the precipitation has been observed between the simulations because the accumulated precipitation is comparable among the runs, but the timing is different (Fig. 8g-i).

## 7 Summary and Conclusions

### 7.1 Summary

During the Alberta Field Project (Thériault et al., 2018), snow was often observed at surface temperatures >0°C at the KES observation site. In general, precipitation occurred during relatively dry conditions. For example, ice-phase precipitation was reported at the surface at temperatures up to 9°C with a relative humidity of 45%. Also, 60% of the particles observed were rimed (Hung, 2017). Given these findings, the relative impact of sublimation and melting of hydrometeors, as well as the role of graupel formation have been investigated. These are addressed by simulating a precipitation event associated with rain and mixed precipitation at the surface, which occurred on 31 March 2015, using the WRF model.

Based on the simulations, a conceptual model explaining the processes leading to the observed precipitation distribution at KES is proposed in Fig. 9. The temperature variations from phase changes impacted the precipitation type, intensity, and

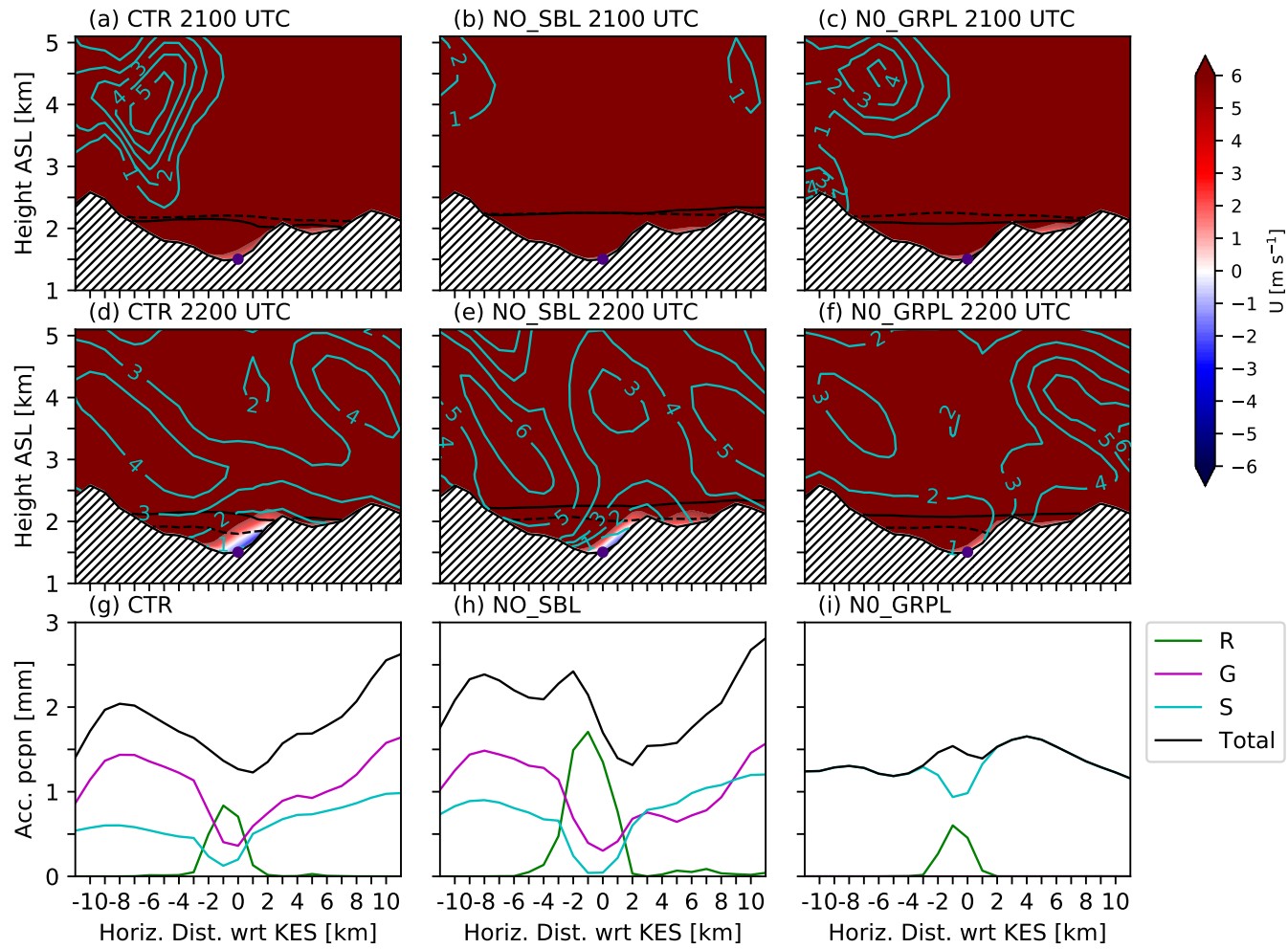

**Figure 8.** Vertical cross section across the Kananaskis Valley showing the snow field (x10 g/m³ blue lines) and zonal wind speed (colour shading) at (a-c) 2100 UTC and (d-f) 2200 UTC as well as (g-i) the accumulated precipitation during the event along the cross-section for (a, d, g) CTR simulation, (b, e, h) NO_SBL simulation and (c, f, i) NO_GRPL simulation. The solid black line indicates the location of the 0°C isotherm at the onset of precipitation and the black dashed line is the 0°C isotherm at the time of the analysis (2100 UTC and 2200 UTC). The shaded area is the topography. The negative (positive) wind speed values are easterly (westerly) winds.

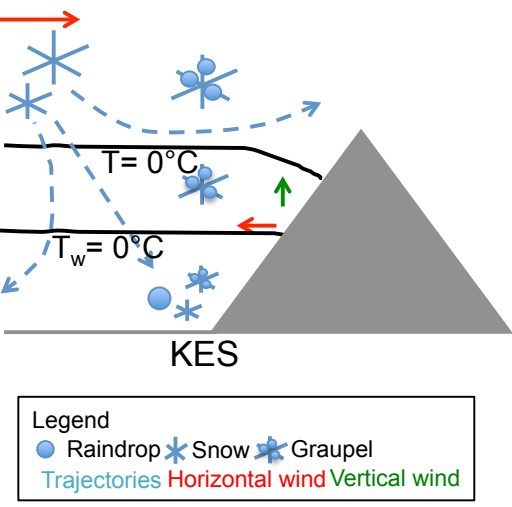

**Figure 9.** Conceptual model explaining the processes driving the evolution of precipitation in the Kananaskis Valley, Alberta. The black solid lines are the 0°C air temperature ($T$) and the wet-bulb temperature ($T_w$). The grey area is the terrain.

its temporal evolution at the surface. The warm conditions of this observed event led to unstable air and resulted in stronger upward motion over a deeper layer. This produced a deep and high ice cloud with liquid water clouds below it. Graupel formed at the top of the liquid cloud and fell rapidly to the surface. At that time, the snow produced on the western barrier is being transported eastward by the wind. The down-valley flow produced by the diabatic cooling from sublimation prevents the snow

from reaching KES because it falls at around 1 m/s. The decrease in mass content is probably associated with a combination of the sublimation of snow and a change in its trajectory associated with the convergence of the flow-field produced by the down-valley flow near the valley floor and the westerly flow aloft. This downslope flow is mainly due to the cold and dense air produced by sublimation. The orographic forcing during the precipitation is weaker because of the strength of the downslope wind.

I It is important to note here that the CTR simulation was rerun with the Thompson et al. (2008) bulk microphysics scheme. This simulation also shows the presence of strong wind shear at KES towards the end of the event. Less snow reached the surface at that time as well (not shown). The results are consistent with our goal to use the model as an analysis tool of physical processes. However, the graupel and snow fields aloft are different as the production of graupel depends strongly on the parameterization of the conversion from snow to graupel and it is different in Thompson et al. (2008) and Milbrandt and Yau

(2005b). First, Thompson et al. (2008) follows Berry and Reinhardt (1974), and Milbrandt and Yau (2005b) follows Murakami (1990), for which the conversion of snow to graupel depends on the collection and the vapor deposition. Second, note that the mass converted into graupel also depends on the assumed size distribution of snow, which is an inverse exponential in Milbrandt and Yau (2005b) but is different in Thompson et al. (2008). In addition, the more recent cloud microphysics scheme called the Predicted Particle Properties (P3, Morrison and Milbrandt, 2015), allows smooth transitions in the riming degree, which

produces a more realistic transition between snow, partially rimed snow and graupel. Finally, relative saturated atmospheric

conditions would lead to a weaker wind shear that could let the snow reach KES. Further investigation should be conducted on this.

The parameterization of graupel formation and evolution could affect the amount and distribution of precipitation at the surface. This study shows that rimed-faster-falling particles and unrimed-slow-falling particles (snow) reaching KES will not be formed at the same location aloft. The CTR produces a small amount of snow at the surface. Given that the conversion to graupel occurred in certain conditions, snow remained aloft longer, which altered the graupel formation and its vertical evolution. This suggests that the amount of graupel maybe underestimated. Even if this is the case, it would not change the physical processes highlighted in Fig. 9 about the sublimation of snow and graupel and the presence of graupel aloft. It can, however, alter the amount of the different types and timing of precipitation reaching the surface depending on the amount of snow conversion into graupel.

## 7.2 Conclusions

Based on the results obtained from the simulations and the conceptual model, key conclusions are as follows.

- The model reproduces well the atmospheric conditions and the precipitation amounts and type.

- Sublimation has a greater impact than melting on the evolution of the precipitation at the surface. This is due to the sub-saturated conditions in the lower atmosphere, which decreases the atmospheric layer where ice-phase precipitation can melt.

- When the thermodynamic impact of sublimation is not considered, it alters the environmental temperature aloft. The warmer conditions create more upward motion, which leads to favourable conditions for accretion (graupel formation) aloft. Furthermore, it allows for a warmer melting layer near the surface resulting in a higher proportion of rain.

- As the precipitation falls and is transported by the wind, it alters the distribution of latent heating due to phase changes and this, in turn, affects the wind direction along the mountainside.

- The trajectories of particles explain some of the differences in the precipitation amounts and types distribution at KES. Snowflakes fall slower than graupel, therefore they tend to more closely follow streamlines. For example, snow reaching the surface at KES is produced on the westward side of the Kananaskis Valley.

- The relative amount of snow reaching KES depends on the strength of the vertical wind shear above KES. Stronger down valley flow will tend to prevent snow particles from reaching KES.

This study has some limitations. First, due to some instrumentation issues, the measurements of wind speed and wind direction during the Alberta Field Project were sometimes inconsistent. Second, this study also has some numerical limitations due to the choice of microphysics parameterization in the WRF and to the specific surface module in WRF for instance. Different microphysics schemes would produce different precipitation rates and thus affect the cooling rate associated with sublimation and melting. In a dry environment with temperatures near 0°C, if snowflakes do not sublimate, it can overestimate the amount

of precipitation produced in models, leading to warm biases. Furthermore, the rate of autoconversion from snow to graupel will also impact the distribution of precipitation aloft and, in turn, at the surface. This is particularly important in complex terrain as previously mentioned in Milbrandt et al. (2009) and Morrison et al. (2015). Using another cloud microphysics scheme, however, should not qualitatively modify results. Similar conclusions on involved physical processes in the distribution and types of

hydrometeors at the surface would be obtained. Other atmospheric conditions should be further investigated. A relatively more saturated environment would lead to different results as, in a case of weak precipitation, a weaker vertical wind shear. In that case, graupel/snow particles do not sublimate and will melt. The diabatic cooling by melting would be reduced, which could allow particles to reach KES. Third, simulations using a particle-tracking model could be used to compute the trajectories of the precipitation particles to better assess the environmental conditions in which they fall.

Overall, this study shows that the microphysical processes leading to precipitation in complex terrain could significantly impact the precipitation intensity and type in the valley. Even if the study is conducted based on a relatively light precipitation event, critical scientific insights on the formation and evolution of precipitation are gained. Accurate representation of precipitation phase changes and accretion leading to graupel, as well as the wind field are critical, in particular in sub-saturated orographic regions such as the eastern slopes of the Canadian Rockies.

*Data availability.*  The dataset used to conduct this study is available upon request to the corresponding author (Julie M. Thériault).These are the Weather and Research Forecasting (WRF) simulations, the necessary information and files to conduct the simulations and the simulations.

*Competing interests.*  No competing interests are present.

*Acknowledgements.*  We would like to thank the Natural Sciences and Engineering Research Council of Canada (NSERC) through the Discovery Grant and the Changing cold regions network (CCRN) as well as Global Water Futures for providing the financial support to

accomplish this work. This study was also supported by a Canadian Research Chair Tier 2. One of the authors (Émilie Poirier) would like to thank the Fond Quebecois de la recherche sur la nature et les technologies (FRQNT) and NSERC for graduate scholarships.

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
