# Peer review of "Role of sublimation and riming in the precipitation distribution in the Kananaskis Valley, Alberta, Canada"

_Hydrology and Earth System Sciences, 2019_

## Referee Comment (RC1) · Anonymous Referee #1 · 19 Mar 2019

General comments:

This manuscript investigates the role of sublimation and riming in orographic precipitation in the Kananaskis Valley based on a well-documented mixed precipitation event from a field complain in the spring 2015. The authors analyzed the observed data and conducted a set of numerical simulations to isolate and quantify the impacts of these two important physical mechanisms in the precipitation process. Their major conclusions include 1) sublimation can have a greater impact than melting on the precipitation evolution under sub-saturated conditions in the lower atmosphere, 2) diabatic cooling due to sublimation or melting can result in change in the precipitation environment,

allowing coupled interactions between orographic flow and precipitation, and 3) the orographic precipitation distribution cannot be simulated adequately if the thermodynamic impact of sublimation (and melting) is not represented correctly in the numerical models.

The data and techniques used in this study are clearly described, referenced, and easy to follow. The conclusions are well-supported and consistent with the stated objectives. This study represents an original and interesting contribution to our understanding of the thermodynamics and microphysics of precipitation in complex terrain. The manuscript is well-organized. But there are some language issues (grammatical and stylistic errors). Some figures need to be revised for clarity. Therefore my recommendation is to accept for publication after some minor revisions.

Specific comments and technical corrections:

The title should be either "Role ... in ..." or "Impact ... on ..."

P1, L9: Replace "where the field campaign took place during March-April 2015" with "during March-April 2015". It has already been mentioned at the beginning that the field campaign took place during this period.

P1, L11: Remove the unnecessary comma after "2015".

P2, L4: You may need to add "which is" before "associated with...".

P2, L8: "the distance associated with complete melting of solid precipitation" may not be considered as a physical mechanism. Isn't it just a factor?

P2, L12: Consider revise the sentence to "However, Zangl (2007) used numerical simulations to demonstrate (or suggest) that..."

P2, L13: I am not sure which event is the "same event".

P2, L14: What do you mean "relatively warm temperature"? It would be better to specify it as "above-freezing temperature".

P2, L28: Consider change the sentence to "precipitation types over Baffin Island, Nunavut, were characterized by Henson et al. (2011) and Fargey et al. (2014)". The study area of Fargey et al. (2014) was not restricted to Iqaluit.

P3: Caption of Fig. 1: Consider also defining those three-letter identifiers with the real location names in the caption.

P3, L7: Remove the comma after "including".

P3, L11: Consider replace the second "during" with "in".

P3, L12: Replace "Thériault et al. (2018)" with "(Thériault et al., 2018)".

P3, L17: "GEONOR" should be defined and referenced here.

P4, Caption of Fig. 2: "CTL" for control should be defined somewhere in the text, and consider change the caption to: "Vertical profiles of air temperature (solid line) and dew point temperature (dashed line) at 2100 UTC 31 March 2015 at the KES site. The measurement and the CTL simulation are represented by blue lines and black lines, respectively."

P4, L10: How do you define bright band in Fig. 3a? Please explain in the text of the figure caption.

P4, L13: Is this 200-m layer a "non-melting layer" or a "partially-melting layer"?

P5, L3, and P17, L10: "WRF" has been defined on P3. You don't need to re-define it here.

P5, L4: Did you "conducted" the 3D simulations, or "used" the simulations conducted by others? The word "used" is confusing.

P6, Section 3.2: About the two-moment microphysics scheme, some recent studies (Morrison et al. 2015, Milbrandt et al. 2016) showed that there is a systematic bias in this scheme, which is linked to the fact that ice-phase particles are represented by
pre-defined categories. Essentially, in situations with light riming, the scheme accounts for the mass growth of snow but not the increase in density and fall speed, unless the riming rate is sufficiently high that snow is converted to graupel, which has a higher terminal fall speed. Such configuration allows lighter hydrometeors to stay in the air too long before being converted to heavier hydrometeors. Could you comment to what extent this bias may affect the simulations in your study?

P7, L4: The acronym "CTL" should be defined earlier, i.e. when it first appears in the text.

P7, L6: Consider changing "latent heating/cooling due to the melting..." to "diabatic heating/cooling due to the precipitation transition". Latent heating is due to the condensation, not from melting of snow.

P7, L20-25: Observations are poorly presented in Fig. 4. See a comment given later (P8, Fig. 4).

P7, L30: You can remove "(<5 knots)". It is kind of confusing. Do you mean the simulated winds are less than 5 knots, or they are not stronger than observed winds for more than 5 knots?

P8, Fig. 4: What do the line contours represent? My guess is elevation. Please mention it in the figure caption. Also, it is hard to read the observations from the circles in (c). It would be better to plot them separately in (d). Or simply mention the observed amounts in the caption.

P11, L2: Change "role" to "roles".

P11, L12: Change "is" to "are".

P11, L15: Do you mean "is considered to produce a similar..."?

P12, L3: Change "suggests" to "suggest".

P12, L6: Change "differs from the CTL simulation" to "differ from their counterparts in

the CTL simulation".

P12, L22: Either delete "studies", or change "cases" to "case".

P13, L6: Replace "changes" with "change".

P17, L14: Do you mean "resulted in stronger upward" (rather than "weak")?

P17, L17: Why are snow particles transported upward due to downslope flow?

P18, L7-10: Operational meteorologists in western Canada noticed that the High-Resolution Deterministic Prediction System (HRDPS) based on the MY2 microphysics scheme often has a warm bias in the valleys. You mentioned on Page 6 that in the MY2 scheme, snow sublimation can only occur when the temperature is below 0C. Based on your conclusion given here, do you think this sublimation restriction is partially responsible for the warm bias?

References:

Milbrandt, J. A., and H. Morrison, 2016: Parameterization of cloud microphysics based on the prediction of bulk ice particle properties. Part III: Introduction of multiple free categories. J. Atmos. Sci., 73, 975–995, doi:10.1175/JAS-D-15-0204.1.

Morrison, H., and J. A.Milbrandt, 2015: Parameterization of cloud microphysics based on the prediction of bulk ice particle properties. Part I: Scheme description and idealized tests. J. Atmos. Sci., 72, 287–311, doi:10.1175/JAS-D-14-0065.1.

---

## Referee Comment (RC2) · Anonymous Referee #2 · 20 Mar 2019

General Comments:

This manuscript describes a numerical modeling study of a weak precipitation event in a mountainous region and examines the importance of the microphysical processes of snow sublimation and riming on the phase and distribution of precipitation at the surface. High resolution (1 km grid spacing) simulations were done with the WRF model using a 2-moment bulk microphysics (MP) scheme. Comparisons were made to local observations, focusing primarily at a single site. Model sensitivity tests were performed whereby specific processes were shut off in the MP scheme and the impacts were

examined. The authors argue that the results illustrate the relative importance of sublimation of snow flakes and snow pellets on altering the temperature at low elevations and thus ultimately the resulting precipitation.

Overall this manuscript is well-written and logically presented, though the figures (regarding the presentation of microphysical fields) need to be improved and reworked (see comments below). The scientific methodology is sound and the conclusions are largely supported by the evidence presented (with some limitations; see below) and provide some understanding of the importance of the processes discussed. As is often the case with studies of this kind that are based on a single case study, the authors need to do a bit more work to illustrate clearly the broader implications of the study. In its present form, the manuscript seems somewhat limited to discussion of the specific details of this specific case. However, this should be straightforward to achieve with some added discussion.

Also, although this is a process study, not an examination of model-specific details, the numerical model – in particular the MP scheme – plays a critical role in the analysis on which all of the scientific conclusions are based. Therefore, I believe that closer examination/discussion of some model details is needed to strengthen the conclusions about the processes and, arguably, to expand the relevance of the conclusions. The manuscript could possibly be published with some improvements to the presentation (see below) and a bit more discussion; however, I think going into some more depth with regards to the MP scheme (see below) could strengthen the paper considerably and I would recommend this approach.

Summary criteria: Scientific significance: 2) good Scientific quality: 2) good Presentation quality: 2) good (but please see specific comments 4 and 5)

Specific Comments:

1. The MP parameterization scheme plays a crucial role in this study. Scientific conclusions are made about the relative roles of sublimation and riming based on what is

simulated by the MP scheme. But due to the complexity of crystal shapes, fall speeds, the (artificial) conversion between snow and graupel (snow pellets), etc., these are difficult processes to model and different schemes parameterize these processes differently. Thus, as presented, the conclusions are weakened by the fact that using a different MP scheme, or even just changing the parameters within the same scheme (with reasonable bounds) could lead to different results. It is not good enough to simply mention that different MP schemes will produce different simulations of sublimation and melting rates (p. 18, line 22) – this point needs to be addressed somehow, either to strengthen the conclusion or to more thoroughly describe the limitations of the results. This is challenging, but it needs to be undertaken to some degree. One idea would be to do some sensitivity tests with changes to the sublimation rates (e.g. changing the capacitance, which is highly simplified in the MP scheme), riming rates (e.g. changing the collection efficiencies for collection of droplets by snow and graupel), rate of conversion between snow and graupel (this is an artificial process anyway), . . . If you can establish that the conclusions are similar despite changes in the parameterization of the process rates within reasonable bounds, this strengthens the conclusions and addresses the inherent limitation regarding the use of a particular MP scheme. If the overall results change dramatically, this is useful in another way in exposing a limitation in this type of modeling study (but you could still make some meaningful comments about the importance of sublimation etc.). Also, some explanation/discussion about how snow and snow pellets, and the processes examined in the study, are are represented in models, and in particular in the specific MP scheme used, should be included.

2. One of the things that comes out of this study is the importance of riming and the impact on the location of precipitation at the surface on whether the rimed ice stays as "snow" or is converted to "snow pellets". As mentioned above, the importance of the "conversion" rate and its parameterization should definitely be included in the discussion, as well as the inherent limitations of an MP scheme that has these abrupt transitions between categories. Also, part of the discussion could include other types of weather cases where the distinction between snow or snow pellets plays a role in

determining the location of precipitation in mountainous region. I am thinking specifically of the IMPROVE-2 study, on which there were several modeling studies using MP schemes. In fact, there were a couple of papers published that used the Milbrandt-Yau MP scheme (Milbrandt et al. 2009, MWR; and Morrison et al. 2016, JAS).

3. The "verification" of the CTR simulation, described in the first few paragraphs of section 4, is a bit weak and should be strengthened. On p.8/ln 20, it states "In summary, the weather conditions at KES are generally well represented by the model." First, I suggest changing this to, "...the meteorology . . . is .. well simulated...". More importantly, you should say generally well represented (simulated) for what purpose, because the simulation is not perfect, as shown in Figs. 2-4. I think what you mean is that it is simulated sufficiently accurately that you can proceed to make meaningful conclusions about your scientific objectives based on the model. This should be stated (and defended). A model reflectivity time series, corresponding to the observations in Fig. 3a), would be useful.

4. It would be useful to have precipitation accumulation maps like Fig. 4 (but with (a) separated into snow and snow pellets as separate panels) for all of the sensitivity runs. Or, perhaps better, for the sensitivity runs plot the differences, EXP(x) – CTR, for each precip type. This would illustrate, e.g., the lateral shifts in precipitation when specific processes are shut off.

5. The presentation of the hydrometeor fields in the figures could be improved considerably. First, linear scales for mixing ratios (or mass contents) do not work well. I suggest hand-picking a few specific ranges for the plotting, and be consistent for all hydrometeor types; e.g.: 1e-6, 1e-5, 1e-4, 2e-4, . . . whatever it takes to clearly illustrate and discriminate low and high values. Explain/show better what is meant by "cloud droplets and rain" (Fig. 5a) – e.g. use different colors (note, rain could be present aloft, formed by coalescence). Also, I suggest plotting mass contents ($\rho_a \ast q_x$), not mixing ratios ($q_x$). For the time series plots, you could combine Fig. 10 with Fig. 5 (i.e. add Fig. 10a panel to Fig. 5), and do this for all runs. This would remove the need for

Figs. 7, 8, and 10, it would provide more info for the sensitivity runs (i.e. magnitudes of values, not present in Fig. 8). This could either be separate 6-panel figures for each run or a single 24-panel, which is probably doable since you would not need to repeat the color legends or y-axes for each run. All this would go along way to improving the presentation and description of the effects of the various sensitivity runs.

Minor Comments:

- In the atmospheric sciences, and certainly in the field of cloud microphysics, the term "graupel" is used. Is there a reason the authors opt to use "snow pellets", which is more of a layperson (or weather forecaster) term? Since this is a scientific article that examines microphysical fields and processes, I would think the authors should use "graupel" throughout, and simply mention briefly early on this graupel is often referred to commonly as "snow pellets" (e.g. in the AMS Glossary of Meteorology).

- p.6/ln 33 – I recommend against making a reference to your M.Sc. Thesis and simply make the claim to these modifications here in this paper. Also, what specifically does the correction to the saturation vapour pressure calculation refer to – was this a bug in the original scheme? (And by chance has it been corrected in any recent official WRF releases?)

- p. 7/ln 8, "accreted particles". Unclear. I assume this means "rimed crystals". Degree of riming? Partially rimed or bona fide graupel?

- Section 3.3: I suggest adding a table of model runs, with the run name and a brief description. For run names, I would suggest (only) "CTR", "NO_MLT", "NO_SUB", and "NO_SNP" (or, better, "NO_GRPL"). For the SNP run, please elaborate on how, specifically, graupel was shut off. The second paragraph ("The data are ...") is not relevant in this section.

- Fig. 3: I suggest adding a panel for model reflectivity, corresponding to panel (a). Also, this and all other discussion/figures about the model simulations are from the

[Figure]

1-km domain only, right? This should be stated clearly somewhere.

- Fig. 4: According to Fig. 3d, a significant amount of the precipitation was from snow pellets, with some snow, at least at that location. Perhaps it would be useful to separate the accumulated precip from snow and snow pellets, rather than combining them in Fig. 4a.

- The time series plots look quite choppy. Is it possible to output the time series with higher temporal frequency in order to produce smoother plots? (Not a big deal; just a suggestion.)

- Was there any "hail" in the simulations? Clearly this case does not support hail in nature, but there is a hail category in the MP scheme which is also used to represent small frozen raindrops. If the model hail mixing ratios are indeed zero in all the simulations, this should be stated and state that for this reason hail is excluded from the figures.

- p. 12, line 6, "Less ice crystals ...[ref to nucleation]. You are not showing number concentration, you are showing mass – there is less ice (crystal category) mass, not fewer number. This is probably not due to nucleation, but rather changes in depositional growth.

- p. 12, line 15, "...more rain reaches the surface because the environmental temperature is higher..." But it is T_wet, not T, that counts (determines melting), right? In that regard, perhaps it would be good to plot the T_wet = 0C isotherm in Fig. 8. (and Fig. 12)

- p. 12, lines 24-26. Suggest omitting paragraph or relocate this as an intro to section 6.

---

## Referee Comment (RC3) · Anonymous Referee #3 · 25 Mar 2019

Review of "Role of sublimation and riming on the precipitation distribution in the Kananaskis Valley, Alberta, Canada" - E. Poirier et al. (2019)

General comments

This manuscript explores the role of sublimation and riming for a weak precipitation event observed in the Canadian Rockies. The study is done with the WRF model using 1 km horizontal grid spacing and a bulk microphysics scheme. The authors made comparisons to data observed at a single site in order to constrain the model simulation. Then, sensitivity tests were performed in order to quantify the impacts of the melting of snow, the sublimation of solid precipitation and the snow pellet formation on

the precipitation features. The main conclusion of this study is that the sublimation can have an important impact on the precipitation evolution in a sub-saturated environment at low elevations.

This manuscript is logically presented and the scientific approach is clear. However, few figures need to be improved (see below). Also, the authors need to add some discussions about the limitations of this study since the comparisons between the model and the observed data are performed at a single site and the conclusions are based on a single case study. Moreover, even if the campaign and the numerical tools are clearly referenced, essential details for this study are missing in the manuscript (see below). The manuscript could be published with some improvements to the presentation and more discussion.

Specific comments

1. This study focuses on the roles of the sublimation, melting and riming processes but details about the microphysics parameterizations used are missing. What are the assumptions used to represent the ice species, the conversion between each species, the terminal velocities. . . then all these assumptions need to be considered in the discussion/explanation of the main results. 2. The local heating/cooling rates associated to the sublimation, the melting and the riming processes are proportional to the mass. It is probably most relevant to plot the mass content of the different species instead of the mixing ratio. Also, the heating/cooling rates can probably be useful to the discussion. You can plot, for example, the vertical profiles of the diabatic heating rate due to microphysics for the different sensitivity tests. 3. The description of the campaign and the available instruments/observations need to be expanded and clarified. For example, Fig 1 shows different sites but the data used in the manuscript were primarily observed at KES. Are there observations available at the other sites? Also, many relevant details for this study are only available in Thériault et al. (2018) and need to be included in this manuscript. It could be interesting to provide a list of the used instruments, the location, the limitations, the observed parameters and the associated references. For

examples, the MMR2 gives the temporal evolution of the vertical profile of the reflectivity and Doppler velocity, and the measurement is affected by the signal attenuation due to e.g. the bright band. Finally, the Parsivel optical disdrometer is mentioned but it is never explained how this instrument is useful. It seems, considering the paper of Thériault et al. (2018), that this instrument is used in order to define the type of the surface precipitation. The different methods (automatic and manual) should be briefly described or at least the authors should specify which one is the most accurate in their opinion.

4. The parameterizations of the microphysics processes evaluated in this study as well as the modifications made to the bulk microphysics scheme of Milbrandt and Yau (2005a,b) should be described in the section 3.2 of the manuscript.

5. The comparison between the CTL simulation and the observations should be discussed in more details, especially the vertical structure. The vertical profile of temperature and dew point temperature obtained in CTL is plotted in Figure2 but never mentioned in the manuscript. The temporal evolution of the vertical profile of the precipitation field observed at KES is given in Fig3a but not compared with the simulation results; at least qualitatively due to the signal attenuation due to the bright band (Matrosov, 2008). MRR2 also provides the Doppler velocity fields; is it possible to compare and assess the species fall speed simulated in the CTL run? Moreover, it is stated several times that the model well reproduces the surface observations. You should say that the CTL simulations reasonably reproduce the observations in order to perform sensitivity studies. However, few parameters simulated in CTL differ from the observations. Indeed, a time shift is visible in the temporal evolution of the accumulated precipitation and the temperature. Do you estimate the impact on the results of this comparison between observations and CTL simulations if you choose another grid box?

6. The figures used to illustrate the sensitivity tests are difficult to interpret. I suggest plotting the differences between the CTL simulations and each sensitivity results. 7. There are spelling and grammar errors throughout the manuscript. I suggest that the

authors read through it carefully and clean it up before resubmitting.

Minor comments

P1-L24. "rain-snow boundary" term is used but defined in the following paragraph. Moreover, the definition is confusing. It is equivalent to the radar bright band/melting layer? P2-L8. Sometimes the term solid precipitation is used. Is it ice-phase precipitation? Or precipitation with high density $\neq$ snow? P2-L12. "using numerical simulations" What type of simulations: 1D, 2D, 3D? P2-L23/28. These studies were performed over mountainous area? P3-Fig1. The title of the right panel is not clear. Do you mean domain area? P3-L11. The name of the field campaign is only given in the conclusions section and should be mentioned here. P4-Fig2. The authors may consider adding details on the skewT-logP diagram in order to define the Lifted Condensation Level and Tw. P4- L9. The relative humidity is never given in the manuscript. If available, the temporal evolution of the relative humidity should be added to the Fig 3. P4-L10. The bright-band is close to the surface at the KES station? P5-L1. "brief period of only snow". According to Fig3e, there is no S period? Fig3. You should increase the y-axis because the reader may have difficulties to extract the values, for example for the temperature. Few elements are missing in the caption: wind barbs definition, hatched region in fig 3a. P6-L5. The boundary conditions forcing is every 3h, 6h or 12h? P6-L7. Add the number of grid points of the innermost domain in order to have an idea of the surface area. P6-L23. The most common term is probably "graupel" instead of "snow pellet". P7-L3/18. The section 3.3 can be summarized in one paragraph because the setup of the sensitivity tests is given twice. Also, the first sentence of the section 4 explains that the CTL simulations will be compared to the available observations described in the previous section. P8-L5. …. also investigated "in the CTL simulations" P8-L10. Fig3e indicates a much shorter period of snow pellet precipitation P8-L11. Indicate on Fig 1. or on Fig 4 where is the cross section plotted on Figures 6, 9 and 11. P8-L15/16. Why the vertical movements would initiate only ice crystals and cloud droplets and not the other species? The amount of snow and snow pellet seem much

larger than the amount of ice crystals and cloud and rain water? What is the role of deposition? Fig.6. The intensity of the vertical wind is difficult to read. Also, the definition of the dashed/solid lines for wind is missing in the caption. The wet bulb temperature is plotted but not mentioned in the Section P11-L14. Why do you choose this threshold in order to plot the Fig8? Fig8 is difficult to interpret; you should make a difference between CTL and each sensitivity test. P12-L8. I do not understand the nucleation citation. You never work with the concentration parameter? P13-L8. "flow reversal". Do you mean wind shear?

---

## Author Comment (AC1) · 30 May 2019

**Responses to Anonymous Referee #1**

**General comments**: This manuscript investigates the role of sublimation and riming in orographic precipitation in the Kananaskis Valley based on a well-documented mixed precipitation event from a field campaign in the spring 2015. The authors analyzed the observed data and conducted a set of numerical simulations to isolate and quantify the impacts of these two important physical mechanisms in the precipitation process. Their major conclusions include 1) sublimation can have a greater impact than melting on the precipitation evolution under subsaturated conditions in the lower atmosphere, 2) diabatic cooling due to sublimation or melting can result in change in the precipitation environment, allowing coupled interactions between orographic flow and precipitation, and 3) the orographic precipitation distribution cannot be simulated adequately if the thermodynamic impact of sublimation (and melting) is not represented correctly in the numerical models.

The data and techniques used in this study are clearly described, referenced, and easy to follow. The conclusions are well-supported and consistent with the stated objectives. This study represents an original and interesting contribution to our understanding of the thermodynamics and microphysics of precipitation in complex terrain. The manuscript is well-organized. But there are some language issues (grammatical and stylistic errors). Some figures need to be revised for clarity. Therefore my recommendation is to accept for publication after some minor revisions.

*We thank Referee #1 for his/her suggestions and comments, which helped improved the manuscript. The manuscript was carefully reread to check for language issues. Comments are addressed point by point below.*

Specific comments and technical corrections:
**Comment 1:** The title should be either "Role ... in ..." or "Impact ... on ..."
*The title is now "Role of sublimation and riming in the precipitation ..."*

**Comment 2:** P1, L9: Replace "where the field campaign took place during March-April 2015" with "during March-April 2015". It has already been mentioned at the beginning that the field campaign took place during this period.
*Correction was made.*

**Comment 3**: P1, L11: Remove the unnecessary comma after "2015".
*This was done.*

**Comment 4**: P2, L4: You may need to add "which is" before "associated with. . .".
*Correction was made.*

**Comment 5**: P2, L8: "the distance associated with complete melting of solid precipitation" may not be considered as a physical mechanism. Isn't it just a factor?

*We agree with the referee. The text was modified as: "These simulations identified two physical mechanisms influencing the location of the rain-snow boundary along the mountainside: cooling by melting of ice-phase particles and adiabatic cooling of rising air. The distance associated with complete melting of ice-phase precipitation was also an important factor."*

**Comment 6:** P2, L12: Consider revise the sentence to "However, Zangl (2007) used numerical simulations to demonstrate (or suggest) that. . ."
*The sentence was revised as: "However, Zängl (2007) used numerical simulations to demonstrate that the cooling..."*

**Comment 7:** P2, L13: I am not sure which event is the "same event".
*It is now mentioned as: "the same event as Steiner et al. (2003)..."*

**Comment 8:** P2, L14: What do you mean "relatively warm temperature"? It would be better to specify it as "above-freezing temperature".
*Yes, it has been replaced.*

**Comment 9:** P2, L28: Consider change the sentence to "precipitation types over Baffin Island, Nunavut, were characterized by Henson et al. (2011) and Fargey et al. (2014)". The study area of Fargey et al. (2014) was not restricted to Iqaluit.
*The sentence was changed to: "In contrast, Henson et al. (2011) and Fargey et al. (2014) characterized precipitation types over Baffin Island, Nunavut, showing that rimed particles, aggregates, and snow pellets were very common even during light precipitation events."*

**Comment 10:** P3: Caption of Fig. 1: Consider also defining those three-letter identifiers with the real location names in the caption.
*They are now defined. The new caption is "Figure 1. Area of interest (left) and 1km mesh domain (right) used for the numerical simulations with the WRF model. BAR stands for the Barrier Lake research station, NAK for Nakiska ski area, KES for the Kananaskis Emergency Services site and FOR for Fortress Mountain. Red line on the right panel indicates the position of the cross section used in Figs. 6, 9 and 11."*

**Comment 11:** P3, L7: Remove the comma after "including". P3, L11: Consider replace the second "during" with "in". P3, L12: Replace "Thériault et al. (2018)" with "(Thériault et al., 2018)".
*These three corrections were made.*

**Comment 12:** P3, L17: "GEONOR" should be defined and referenced here.
*The new sentence reads as follows: "Instrumentation used included a GEONOR weighing precipitation gauge (Rasmussen et al., 2012), ...".*

**Comment 13:** P4, Caption of Fig. 2: "CTL" for control should be defined somewhere in the text, and consider change the caption to: "Vertical profiles of air temperature (solid line) and dew point temperature (dashed line) at 2100 UTC 31 March 2015 at the KES site. The measurement and the CTL simulation are represented by blue lines and black lines, respectively."

*The caption of Fig. 2 was changed to "Vertical profiles of air temperature (T, solid line), dew point temperature (Td, dashed line) and wet-bulb temperature (Tw, light colour) at 2100 UTC 31 March 2015 at the KES site. The measurement (OBS) and the control simulation (model) described in section 3.1, are represented by blue and black/grey lines, respectively."*

**Comment 14:** P4, L10: How do you define bright band in Fig. 3a? Please explain in the text of the figure caption.

*The bright band is now defined in the 2ⁿᵈ paragraph of the introduction. It reads as follows "… , which is associated with a maximum reflectivity value (> 30 dBZ) called the radar bright band (Fabry and Zawadzki, 1995)." Given that the sentence related to Fig. 3a reads as follows "The radar reflectivity bright band (>30 dBZ) is located at the elevation where ice-phase precipitation started to melt (Fig. 3a)". The following sentence was also added to the figure caption. "Reflectivity values > 30 dBZ are associated with the radar reflectivity bright-band.".*

*This is the revised Fig. 3.*

[Figure]

*Figure 3 (revised): Atmospheric conditions and precipitation fields during the 31 March 2015 event at KES. (a) Reflectivity field measured by the Micro Rain Radar and (b) is estimated by the model (CTR). Reflectivity values > 30dBZ are associated with the radar reflectivity bright-band.;*

*(c) surface temperature (T) and relative humidity (RH) observed (black line) and simulated (blue line); (d) wind speed and direction using wind barbs, where the observed is black and simulated is blue. An empty circle is wind speed rounded at 0 knots, a short bar is rounded at 5 knots; (e) unadjusted liquid equivalent accumulated precipitation observed (black line, OBS) and simulated (bold blue line for total, green line for rain, thin blue line for graupel and dashed blue line for snow), and (f) the type of precipitation observed manual (MAN) and automatically (AUT) at KES. These are rain (R), graupel (GR), snow (S), mixed precipitation (M), heavily rimed snow (HR), rimed aggregates (RA), dry snow (DS) and dendrites (DE). Simulated results are for the CTL run. Adapted from Thériault et al. (2018).*

**Comment 15:** P4, L13: Is this 200-m layer a "non-melting layer" or a "partially-melting layer"?
*This layer was associated with only solid precipitation, so a 'non-melting layer'. The sentence was revised to "The rain-snow transition is located about 200 m below the 0°C isotherm, which confirms that solid precipitation was not melting until the level associated with a wet-bulb temperature, $T_W$, > 0°C was reached (Harder and Pomeroy, 2013). ".*

**Comment 16:** P5, L3, and P17, L10: "WRF" has been defined on P3. You don't need to re-define it here.
*Correction was made.*

**Comment 17:** P5, L4: Did you "conducted" the 3D simulations, or "used" the simulations conducted by others? The word "used" is confusing.
*We conducted the simulations. The first two sentences of section 3.1 were changed to "Three-dimensional (3D) simulations are performed using WRF model, version 3.7.1 (Skamarock and Klemp, 2008), with initial and boundary conditions provided by the North American Regional Reanalysis (NARR) data from the National Center for Environmental Prediction (NCEP) (Mesinger et al., 2006).".*

**Comment 18:** P6, Section 3.2: About the two-moment microphysics scheme, some recent studies (Morrison et al. 2015, Milbrandt et al. 2016) showed that there is a systematic bias in this scheme, which is linked to the fact that ice-phase particles are represented by pre-defined categories. Essentially, in situations with light riming, the scheme accounts for the mass growth of snow but not the increase in density and fall speed, unless the riming rate is sufficiently high that snow is converted to graupel, which has a higher terminal fall speed. Such configuration allows lighter hydrometeors to stay in the air too long before being converted to heavier hydrometeors. Could you comment to what extent this bias may affect the simulations in your study?
*Three comments were added to the manuscript to discuss this:*
*1) A comment about P3 was added at the end of section 7.1. It reads as follows. "... Finally, the Predicted Particle Properties (P3, Morrison and Milbrandt 2015 and Milbrandt and Morrison, 2016) allows smooth transitions in the riming degree, which produces a more realistic transition between snow, partially rimed snow and graupel."*

*2) A new paragraph was added at the end of section 7.1 (after the previous answer): "The parameterization of graupel formation and evolution could affect the amount and distribution of precipitation at the surface. This study shows that rimed-faster-falling particles and unrimed-slow-falling particles (snow) reaching KES will not be formed at the same location aloft and it depends strongly on the parameterization. For example, the CTR produces a small amount of snow at the surface. Given that the conversion to graupel occurred in certain conditions, snow remained aloft longer, which altered the graupel formation and its vertical evolution. This suggests that the amount of graupel may be underestimated. Even if this is the case, it would not change the physical processes highlighted in Fig. 12 about the sublimation of snow and graupel and the presence of graupel aloft. It can, however, alter the amount of the different types and timing of precipitation reaching the surface depending on the amount of snow conversion into graupel."*

*3) Another comment was also added to the conclusion (section 7.2): "Different microphysics schemes would produce different precipitation rates and thus affect the cooling rate associated with sublimation and melting. In a dry environment with temperatures near 0°C, if snowflakes do not sublimate it can overestimate the amount of precipitation produced in models, leading to warm biases. Furthermore, the rate of autoconversion from snow to graupel will also impact the distribution of precipitation aloft and, in turn, at the surface. This is particularly important in complex terrain as previously mentioned in Milbrandt et al. (2009). Using another cloud microphysics scheme, however, should not qualitatively modify results. Similar conclusions on involved physical processes in the distribution and types of hydrometeors at the surface would be obtained."*

**Comment 19:** P7, L4: The acronym "CTL" should be defined earlier, i.e. when it first appears in the text.
*The first occurrence in the text of CTR is now in section 3.1 where it is defined.*

**Comment 20:** P7, L6: Consider changing "latent heating/cooling due to the melting. . ." to "diabatic heating/cooling due to the precipitation transition". Latent heating is due to the condensation, not from melting of snow.
*This section was updated and we used diabatic heating/cooling instead of latent heating/cooling.*

**Comment 21:** P7, L20-25: Observations are poorly presented in Fig. 4. See a comment given later (P8, Fig. 4).
*See reply to comment 23 below. The amount for each station was added to the caption.*

**Comment 22:** P7, L30: You can remove "(<5 knots)". It is kind of confusing. Do you mean the simulated winds are less than 5 knots, or they are not stronger than observed winds for more than 5 knots?

*It has been removed. The winds simulated are stronger during the events (~5 knots). It has been clarified in the text and in the figure caption.*

**Comment 23:** P8, Fig. 4: What do the line contours represent? My guess is elevation. Please mention it in the figure caption. Also, it is hard to read the observations from the circles in (c). It would be better to plot them separately in (d). Or simply mention the observed amounts in the caption.

*It is now mentioned in the caption of Fig. 4: "Line contours represent the topography.".*
*We think that it is better to show a direct comparison between observed and simulated accumulated precipitation with circles in (c). Circles are now larger. The numbers are added in the caption as suggested. The numbers are KES (2.7 mm), Nakiska (2.2 mm), Fortress (3 mm) and Barrier Lake Station (0.8 mm). We hope that it is now clearer.*
*This is the revised Fig. 4.*

[Figure]

*Figure 4 (revised): Simulated unadjusted accumulated solid precipitation (mm) including (a) snow and (b) graupel, (c) rain and (d) total accumulated precipitation between 2000 UTC 31 March 2015 and 0000 UTC 1 April 2015. The coloured circles in (d) are the observations at 4 locations. These are KES (2.7 mm), Nakiska (2.2 mm), Fortress (3.0 mm) and Barrier Lake Station (0.8 mm). Accumulated precipitation is in liquid equivalent. The black lines are the topography in meters.*

**Comment 24:** P11, L2: Change "role" to "roles".
*The correction was made.*

**Comment 25:** P11, L12: Change "is" to "are".
*It has been changed.*

**Comment 26:** P11, L15: Do you mean "is considered to produce a similar. . ."?

*The sentence was changed to "The distribution of hydrometeors at KES for NO_MLT is similar to the CTR with very little change in precipitation and cloud distribution (Fig. 8a and c, which are now in the revised Fig. 5 – 1ˢᵗ and 2ⁿᵈ column)."*
*This is the revised Fig. 5.*

[Figure]

*Figure 5 (revised): Comparison of the time evolution of hydrometeors at the surface and aloft at KES during the 4 simulations conducted for CTR, NO_MLT, NO_SBL and NO_GRPL from left to right. (a-d) is ice mass content (x10 g/kg), (e-h) is clouds and rain mass content, where rain is only formed through melting of ice, so it is only present near the surface, (i-l) is graupel mass content, (m-p) is snow mass content and (q-t) is the surface precipitation rate of rain (R), graupel (G) and snow (S). The 0°C isotherm is indicated by the solid black line on (a-p). Panels a-p have the same colour scale.*

**Comment 27:** P12, L3: Change "suggests" to "suggest".
*It has been changed.*

**Comment 28:** P12, L6: Change "differs from the CTL simulation" to "differ from their counterparts in the CTL simulation".
*It has been changed.*

**Comment 29:** P12, L22: Either delete "studies", or change "cases" to "case".
*The word 'studies' was deleted.*

**Comment 30:** P13, L6: Replace "changes" with "change".
*The correction was made.*

**Comment 31:** P17, L14: Do you mean "resulted in stronger upward" (rather than "weak")?
*It should be "stronger" instead of "weaker" (shown in Fig. 10). It is corrected in the text.*

**Comment 32:** P17, L17: Why are snow particles transported upward due to downslope flow?
*The sentences starting on P17, L16 to L18 were clarified as follows. "The snowflakes produced on the western barrier are being transported eastward by the wind. The down valley flow produced by the diabatic cooling from sublimation prevents the snow from reaching KES because it falls at around 1 m/s. The decrease in mass content is probably associated with a combination of the sublimation of snow and a change in its trajectory associated with the convergence of the flow field produced by the down valley flow near the valley floor and the westerly flow aloft."*

**Comment 33:** P18, L7-10: Operational meteorologists in western Canada noticed that the High-Resolution Deterministic Prediction System (HRDPS) based on the MY2 microphysics scheme often has a warm bias in the valleys. You mentioned on Page 6 that in the MY2 scheme, snow sublimation can only occur when the temperature is below 0C. Based on your conclusion given here, do you think this sublimation restriction is partially responsible for the warm bias?
*This restriction could partly explain this warm bias of HRDPS observed in the valley because allowing snow sublimation at temperature above 0°C produces cold and dense air locally in the valley. A comment was added in the conclusion: "In a dry environment with temperatures near 0°C, if snowflakes do not sublimate it can overestimate the amount of precipitation produced in models that lead to warm biases."*

---

## Author Comment (AC2) · 30 May 2019

**Response to Anonymous Referee #2**
**General comments**: This manuscript describes a numerical modeling study of a weak precipitation event in a mountainous region and examines the importance of the microphysical processes of snow sublimation and riming on the phase and distribution of precipitation at the surface. High-resolution (1 km grid spacing) simulations were done with the WRF model using a 2-moment bulk microphysics (MP) scheme. Comparisons were made to local observations, focusing primarily at a single site. Model sensitivity tests were performed whereby specific processes were shut off in the MP scheme and the impacts were examined. The authors argue that the results illustrate the relative importance of sublimation of snow flakes and snow pellets on altering the temperature at low elevations and thus ultimately the resulting precipitation.

Overall this manuscript is well-written and logically presented, though the figures (regarding the presentation of microphysical fields) need to be improved and reworked (see comments below). The scientific methodology is sound and the conclusions are largely supported by the evidence presented (with some limitations; see below) and provide some understanding of the importance of the processes discussed. As is often the case with studies of this kind that are based on a single case study, the authors need to do a bit more work to illustrate clearly the broader implications of the study. In its present form, the manuscript seems somewhat limited to discussion of the specific details of this specific case. However, this should be straightforward to achieve with some added discussion. Also, although this is a process study, not an examination of model-specific details, the numerical model – in particular the MP scheme – plays a critical role in the analysis on which all of the scientific conclusions are based. Therefore, I believe that closer examination/discussion of some model details is needed to strengthen the conclusions about the processes and, arguably, to expand the relevance of the conclusions. The manuscript could possibly be published with some improvements to the presentation (see below) and a bit more discussion; however, I think going into some more depth with regards to the MP scheme (see below) could strengthen the paper considerably and I would recommend this approach.

*We thank Referee #2 for his/her suggestions and comments, which helped improve the manuscript.*

*We agree with the referee that the aim of our study is to identify physical processes leading to the observed phase and distribution of precipitation at the surface in specific conditions i.e. dry sub-cloud layer and mountainous area. In this context, we think that, after demonstrating the ability of the model using the particular Milbrandt and Yau (2005a) MP scheme to simulate the observed case study, our approach using sensitivity experiments is valuable to identify these physical processes regardless of the MP scheme used. We hope that it is now clearer in the introduction of the paper as detailed below in the answer to specific comment #1.*

*Specific and minor comments are now addressed point by point below.*

Specific comments:
**Comment 1**: The MP parameterization scheme plays a crucial role in this study. Scientific

conclusions are made about the relative roles of sublimation and riming based on what is simulated by the MP scheme. But due to the complexity of crystal shapes, fall speeds, the (artificial) conversion between snow and graupel (snow pellets), etc., these are difficult processes to model and different schemes parameterize these processes differently. Thus, as presented, the conclusions are weakened by the fact that using a different MP scheme, or even just changing the parameters within the same scheme (with reasonable bounds) could lead to different results. It is not good enough to simply mention that different MP schemes will produce different simulations of sublimation and melting rates (p. 18, line 22) – this point needs to be addressed somehow, either to strengthen the conclusion or to more thoroughly describe the limitations of the results. This is challenging, but it needs to be undertaken to some degree. One idea would be to do some sensitivity tests with changes to the sublimation rates (e.g. changing the capacitance, which is highly simplified in the MP scheme), riming rates (e.g. changing the collection efficiencies for collection of droplets by snow and graupel), rate of conversion between snow and graupel (this is an artificial process anyway), . . . If you can establish that the conclusions are similar despite changes in the parameterization of the process rates within reasonable bounds, this strengthens the conclusions and addresses the inherent limitation regarding the use of a particular MP scheme. If the overall results change dramatically, this is useful in another way in exposing a limitation in this type of modeling study (but you could still make some meaningful comments about the importance of sublimation etc.). Also, some explanation/discussion about how snow and snow pellets, and the processes examined in the study, are represented in models, and in particular in the specific MP scheme used, should be included.

*We addressed this comment in 3 steps.*

1) *Clarifying the goal of the study. The goal of our study is to identify key physical processes that are associated with the distribution of precipitation types in the Kananaskis valley. We are confident that the physical processes identified in our study are not dependent to the MP scheme used as long as graupel is parameterized. We are mainly interested in studying the impact of the presence of graupel on the distribution of precipitation. The detailed microphysical mechanisms leading to graupel and how they compare in different schemes could be conducted in future work. We tried to put forward the use of the numerical simulations as an analysis tool for physical processes. We hope that it is now clearer.*

   a. *First, this is now clarified in the introduction with this added sentence at the end of 6th paragraph: "After verifying that the model is able to represent this observed case study, numerical simulations are used to investigate physical processes producing the distribution of precipitation in the Kananaskis area.".*

   b. *Second, in the summary part, the following statement has been added at the end of the section: "It is important to notice here that the CTR simulation was rerun with the Thompson et al. (2008) cloud microphysics scheme. This simulation also shows the presence of strong wind shear at KES towards the end of the event.* Less snow reached the surface at that time as well (not shown). *The results are consistent with our goal to use the model as an analysis tool of physical*

*processes.". Note some minor modifications to the conceptual model (Fig. 12) were applied.*

    *c.  Third, the following sentence has been added in the conclusion in the paragraph about the limitations of our study: "Different microphysics schemes would produce different precipitation rates and thus affect the cooling rate associated with sublimation and melting. In a dry environment with temperatures near 0°C, if snowflakes do not sublimate it can overestimate the amount of precipitation produced in models leading to warm biases. However, as highlighted above in the summary, using another cloud microphysics scheme should not qualitatively modify results. Similar conclusions on involved physical processes in the distribution and types of hydrometeors at the surface would probably be obtained. Other atmospheric conditions should be further investigated. Relatively more saturated environment would lead to different results as, in a case of weak precipitation, a weaker vertical wind shear. In that case, solid particles do not sublimate and will melt. The diabatic cooling by melting would be reduced, which could allow particles to reach KES.".*

2) *A short description of the graupel formation was added to section 3.2. "The two-moment microphysics scheme predicts the mass mixing ratio and the total number concentration of inverse exponential size distribution of six hydrometeor categories: cloud droplets, rain, ice crystals, snow, graupel and hail. Each category is described by an assumed mass-diameter relationship and an associated fall speed. The evolution of clouds and precipitation is based on many microphysical processes that are mainly divided into cold and warm processes in the microphysics scheme. In this study we focus on the sublimation and melting of ice, snow and graupel as well as the impact of the presence of graupel. This last process includes the collision/coalescence of ice crystals and snow with cloud droplets or raindrops leading to rimed particles. This parameterization differs among bulk microphysics scheme. For example, Milbrandt and Yau (2005) follows Murakami (1990) to parameterize the conversion of snow-graupel. It is based on the rate of collection of snow/ice with cloud droplets as well as vapor deposition. The change from the snow category to graupel category involves a sharp increase in density (100 to 400 kg/m3) and, in turn an increase in the fall velocity (~1 to 3 m/s). Hence, the mass of snow can increase aloft without falling faster until it is converted into graupel. Pre-defined hydrometeor categories are a limitation of bulk microphysics schemes. A more detail description of the conversion process as well as all processes are given in Milbrandt and Yau (2005a, b)."*

3) *Test with another microphysical scheme. To show that the change in the vertical wind shear at KES, which impacts the distribution of precipitation at the surface, the CTR was run with the Thompson scheme (Thompson et al., 2008). For instance, the figure below is the same as Fig. 11 of the original manuscript with the CTR run with Thompson et al. (2008) (panels c and d). We see clearly that there is a strong vertical wind shear above KES. The strength is different than CTR because different amount of snow and graupel is*

*produced but solid precipitation is still falling in sub-saturated conditions.*

[Figure]

*Figure A: The time evolution of the snow field (mass content, g/kg x 10) and the horizontal (east-west) wind field. The black line is the 0°C isotherm at the onset of the event and the dashed black line is at the time indicated on the panel. (a-b) is CTR and (c-d) is with THOM. This figure is only to illustrate the response to the referee and will not be added to the manuscript.*

**Comment 2**: One of the things that comes out of this study is the importance of riming and the impact on the location of precipitation at the surface on whether the rimed ice stays as "snow" or is converted to "snow pellets". As mentioned above, the importance of the "conversion" rate and its parameterization should definitely be included in the discussion, as well as the inherent limitations of an MP scheme that has these abrupt transitions between categories. Also, part of the discussion could include other types of weather cases where the distinction between snow or snow pellets plays a role in determining the location of precipitation in mountainous region. I am thinking specifically of the IMPROVE-2 study, on which there were several modeling studies using MP schemes. In fact, there were a couple of papers published that used the Milbrandt-Yau MP scheme (Milbrandt et al. 2009, MWR; and Morrison et al. 2016, JAS).

*Yes, the parameterization of graupel is very sensitive to the amount of snow converted into graupel. Some details on the processed studied (graupel formation, melting, sublimation) are now given in section 3.2 as follows. "The two-moment microphysics scheme predicts the mass mixing ratio and the total number concentration of inverse exponential size distribution of six hydrometeor categories: cloud droplets, rain, ice crystals, snow, graupel (i.e. graupel in this study), and hail. Each category is described by an assumed mass-diameter relationship and an associated fall speed. The evolution of clouds and precipitation is based on many microphysical processes that are mainly divided into cold and warm processes in the microphysics scheme. In this study we focus on the sublimation and melting of ice, snow and graupel as well as the impact of the presence of graupel. This last process includes the collision/coalescence of ice crystals and*

*snow with cloud droplets or raindrops leading to rimed particles. Milbrandt and Yau (2005) follows Murakami (1990) to parameterize the conversion snow-graupel. It is based on the rate of collection of snow/ice with cloud droplets as well as vapor deposition. The change from the snow category to graupel category involves a sharp increase in density (100 to 400 kg/m$^3$) and, in turn, an increase in the fall velocity (~1 to 3 m/s). The definition of specific hydrometeor categories is a limitation of bulk microphysics scheme. A more detail description of the conversion process as well as all processes is given in Milbrandt and Yau (2005a, b)."*

*Also, a short discussion has been added at the end of section 7.1. It reads as follows: "the graupel and snow fields aloft are different as the production of graupel depends strongly on the parameterization of the conversion from snow to graupel and it is different in Thompson et al. (2008) and Milbrandt and Yau (2005). First, Thompson et al. (2008) follows Berry et al. (1974), and Milbrandt and Yau (2005) follows Murakami (1990). The latter depends on the collection and the vapor deposition. Second, note that the mass converted into graupel also depends on the assumed size distribution of snow, which is an inverse exponential in Milbrandt and Yau (2005) but is different in Thompson et al. (2008). Finally, the Predicted Particle Properties (P3, Morrison and Milbrandt 2015; Morrison et al., 2016) allows smooth transitions in the riming degree, which produces a more realistic transition between snow, partially rimed snow and graupel. "*

*Finally, a comment on the importance on the conversion snow to graupel in complex terrain is mentioned in the conclusion. It refers to Milbrandt et al. 2009 and Morrison et al. 2016, JAS.*

**Comment 3**: The "verification" of the CTR simulation, described in the first few paragraphs of section 4, is a bit weak and should be strengthened. On p.8/ln 20, it states "In summary, the weather conditions at KES are generally well represented by the model." First, I suggest changing this to, "...the meteorology . . . is .. well simulated...". More importantly, you should say generally well represented (simulated) for what purpose, because the simulation is not perfect, as shown in Figs. 2-4. I think what you mean is that it is simulated sufficiently accurately that you can proceed to make meaningful conclusions about your scientific objectives based on the model. This should be stated (and defended). A model reflectivity time series, corresponding to the observations in Fig. 3a), would be useful.

*This part has been completely rewritten and separated into two sub-sections, one comparing CTR results with observations at KES and another one analyzing the vertical distribution of hydrometeors simulated by the CTR run at KES. The simulated reflectivity time series has been added to Fig. 3 as panel (b).*

[revised manuscript text omitted]

**Comment 4**: It would be useful to have precipitation accumulation maps like Fig. 4 (but with (a) separated into snow and snow pellets as separate panels) for all of the sensitivity runs. Or, perhaps better, for the sensitivity runs plot the differences, EXP(x) – CTR, for each precip type. This would illustrate, e.g., the lateral shifts in precipitation when specific processes are shut off.

*Snow and graupel have been separated in a new version of Figure 4. Also, NO_SUB and NO_GRPL have been compared with the control run in the figures (Figures B and C) below. Note that there is not necessarily a lateral shift in precipitation, because the accumulated precipitation is similar in all cases. These are 1.64 mm CTR, 2.18 mm for NO_MLT, 1.39 mm for NO_SBL and 1.45 mm for NO_GRPL. There is, however, a change in the timing of the precipitation intensity depending on the sensitivity experiment. It has been clarified in the text at the end of section 6. "Note that no lateral shift of the precipitation has been observed between the simulations because the accumulated precipitation is comparable among the runs, but the timing is different."*

[Figure]

*Figure 4 (revised): Simulated unadjusted accumulated solid precipitation (mm) including (a) snow and (b) graupel, (c) rain and (d) total accumulated precipitation between 2000 UTC 31 March 2015 and 0000 UTC 1 April 2015. The coloured circles in (d) are the observations at 4 locations. These are KES (2.7 mm), Nakiska (2.2 mm), Fortress (3.0 mm) and Barrier Lake Station (0.8 mm). Accumulated precipitation is in liquid equivalent. The black lines are the topography in meters.*

[Figure]

*Figure B: The simulations without the temperature change from sublimation (NO_SBL) compared with CTR (NO_SBL-CTR).*

*Figure C: The simulations without the temperature change from sublimation (NO_GRPL) compared with CTR (NO_GRPL-CTR).*

**Comment 5**: The presentation of the hydrometeor fields in the figures could be improved considerably. First, linear scales for mixing ratios (or mass contents) do not work well. I suggest hand-picking a few specific ranges for the plotting, and be consistent for all hydrometeor types; e.g.: 1e-6, 1e-5, 1e-4, 2e-4, . . . whatever it takes to clearly illustrate and discriminate low and high values. Explain/show better what is meant by "cloud droplets and rain" (Fig. 5a) – e.g. use different colors (note, rain could be present aloft, formed by coalescence). Also, I suggest plotting mass contents (rho_a*q_x), not mixing ratios (q_x). For the time series plots, you could combine Fig. 10 with Fig. 5 (i.e. add Fig. 10a panel to Fig. 5), and do this for all runs. This would remove the need for Figs. 7, 8, and 10, it would provide more info for the sensitivity runs (i.e. magnitudes of values, not present in Fig. 8). This could either be separate 6-panel figures for each run or a single 24-panel, which is probably doable since you would not need to repeat the color legends or y-axes for each run. All this would go along way to improving the presentation and description of the effects of the various sensitivity runs.

*The figures have been redone mainly as suggested. The main changes are:*

*1)  Figure 5 is a 18-panel figure showing all 4 experiments. With this revised Figs. 5 and 6, we deleted Figs. 7, 8 and 9 but kept Figs. 10 and 11.*

[Figure]

*Figure 5 (revised): Comparison of the time evolution of hydrometeors at the surface and aloft at KES during the 4 simulations conducted for CTR, NO_MLT, NO_SBL and NO_GRPL from left to right. (a-d) is ice mass content (x10 g/kg), (e-h) is clouds and rain mass content, where rain is only formed through melting of ice, so it is only present near the surface, (i-l) is graupel mass content, (m-p) is snow mass content and (q-t) is the surface precipitation rate of rain (R), graupel (G) and snow (S). The 0°C isotherm is indicated by the solid black line on (a-p). Panels a-p have the same colour scale.*

*2) Figure 6 is similar as Fig. 5 but for the vertical cross-section.*

[Figure]

*Figure 6 (revised): Comparison of the vertical cross-section across the Kananaskis Valley along the red line in Fig. 1 showing the mass content of hydrometeors during the 4 simulations conducted for CTR, NO_MLT, NO_SBL and NO_GRPL from left to right. (a-d) is ice mass content (x10 g/kg) with vertical velocity (m/s). The yellow line is 0 m/s, the dashed lines are negative values and solid lines are positive values, (e-h) is clouds and rain mass content, (i-l) is graupel mass content and (m-p) is snow mass content. The 0°C isotherm is indicated by the solid black line. Panels a-p have the same colour scale. The location of KES is indicated by the purple dot.*

*3) In this case we deleted Figs. 7-9 but kept Fig. 10 (plotted mass content instead of mass mixing ratio).*

*4) Clouds and rain were kept on the same panel because rain is only located below the 0°C isotherm and the clouds are aloft. There is no rain aloft in those experiments. A comment was added to the text.*

*5) The temperature fields were not added to the new Figs. 5 and 6 because it was too busy so we decided to keep Fig. 10. However, so you can see the temperature fields in detail, we added a 4-panel figure only in the responses:*

[Figure]

*Figure D: Shows the (a) temperature fields time-series at KES for the CTR, (b) NO_SBL-CTR, (c) NO_MLT-CTR and (d) NO_GRPL-CTR. We see most of the difference between NO_SBL-CTR, which is the reason why we kept Fig. 10 in the manuscript.*

Minor Comments:

**Comment 1**: In the atmospheric sciences, and certainly in the field of cloud microphysics, the term "graupel" is used. Is there a reason the authors opt to use "snow pellets", which is more of a layperson (or weather forecaster) term? Since this is a scientific article that examines microphysical fields and processes, I would think the authors should use "graupel" throughout, and simply mention briefly early on this graupel is often referred to commonly as "snow pellets" (e.g. in the AMS Glossary of Meteorology).

*The term "snow pellets" has been changed to graupel throughout the text.*

**Comment 2**: p.6/ln 33 – I recommend against making a reference to your M.Sc. Thesis and simply make the claim to these modifications here in this paper. Also, what specifically does the correction to the saturation vapour pressure calculation refer to – was this a bug in the original scheme? (And by chance has it been corrected in any recent official WRF releases?)

*The modifications made on the original Milbrandt and Yau (2005a) scheme are now detailed in the paper. The bug has been corrected into more recent versions of WRF. Section 2 was updated to: "Given that graupel can sublimate at temperatures >0°C, the same equation was used for snow, which is*

$$QVD_{vs} = \frac{1}{AB_i}\left[\, 2\pi(S_i - 1)N_{0s}VENT_s - \frac{L_sL_f}{K_aR_vT^2}QCL_{cs} \,\right]$$

*where*

$$AB_i = \frac{L_s^2}{K_aR_vT^2} + \frac{1}{\rho q_{is}\psi}$$

*is the thermodynamic function. Also, $S_i$ is the saturation ratio with respect to ice, $N_{0S}$ is the intercept parameter for snow, $VENT_S$ is the mass-weighted ventilation factor (Ferrier, 1994), $K_a$ is the thermal conductivity of air, $R_v$ is the gas constant for water vapour, $T$ is the temperature of air, $\rho$ is the density of air, $q_{is}$ is the saturation vapour mixing ratio with respect to ice and $\psi$ is the diffusivity of water vapour in air.*
*The sublimation rate equation was moved in the microphysics scheme so that snow and graupel sublimation are computed in the same conditions, at all air temperatures. The function polysvp was also corrected in the microphysics scheme to calculate the saturation vapour pressure properly at all temperatures. This bug was fixed in the following version of WRF."*

**Comment 3**: p. 7/ln 8, "accreted particles". Unclear. I assume this means "rimed crystals". Degree of riming? Partially rimed or bona fide graupel?
*The term "accreted particles" has been changed to "rimed particles".*

**Comment 4**: Section 3.3: I suggest adding a table of model runs, with the run name and a brief description. For run names, I would suggest (only) "CTR", "NO_MLT", "NO_SUB", and "NO_SNP" (or, better, "NO_GRPL"). For the SNP run, please elaborate on how, specifically, graupel was shut off. The second paragraph ("The data are . . .") is not relevant in this section.
*Since we only have 3 experiments; we described them in bullet points in section 3.3. The name of the runs have been changed throughout the text as suggested. The details on how the graupel was shut off have been added as "The control simulation (CTR) is conducted using the modified microphysics and model configuration described in section 3.1 5 and 3.2. To estimate the impact of temperature changes while neglecting the diabatic heating/cooling due to the precipitation phase transition and no graupel formation. The temperature tendency equation is*

$$\frac{dT}{dt} = \frac{1}{\Delta t}\left\{ \frac{L_f}{c_{pd}}\left(\begin{array}{c} \Delta QCLcs + \Delta QCLcg + \Delta QCLch + \Delta QCLri + \Delta QCLrs \\ +\Delta QCLrg + \Delta QCLrh + \Delta QFZci + \Delta QFZrh \\ -\Delta QMLir - \Delta QMLsr - \Delta QMLgr - \Delta QMLhr \end{array}\right) \right. \\ \left. + \frac{L_s}{c_{pd}}\left(\begin{array}{c} \Delta QNUvi + \Delta QVDvi + \Delta QVDvs + \Delta QVDvg \\ +\Delta QVDvh \end{array}\right) \right\}$$

*where $L_f$ is the latent heat of fusion, $L_S$ is the latent heat of sublimation, $c_{pd}$ is the specific heat of dry air and Q is for mixing ratio. The types of mixing ratios are noted by CL for collection, FZ for freezing, ML for melting, NU for nucleation, VD for diffusional growth (positive) or*

*sublimation (negative) and the subscripts (c, r, i, s, g, h, v) represent cloud droplets, rain, ice, snow, graupel, hail and water vapour.*

*The three key following sensitivity experiments were performed:*

> *1.     NO_MLT: The diabatic cooling of melting snow (QMLsr) and graupel (QMSLgr) were set to zero in the temperature tendency equation. Hence, snow and graupel were allowed to melt into rain but no energy was extracted from the environment to melt the particles.*

> *2.     NO_SBL: The diabatic cooling of sublimation of snow (QVDvs) and graupel (QVDvg) was set to zero in the temperature tendency equation.*

> *3.     NO_GRPL: Since graupel was often reaching the surface at KES during the Alberta field project (Thériault et al., 2018), another simulation was performed. The initiation of graupel was suppressed by turning the production of graupel off (grpl _ON = false) . It was also ensured that there were no sources or sinks, hence, no warming from the cloud droplets freezing on the solid particles (snow or/and ice) and no sublimation of graupel since none was produced."*

*A new section, section 3.4 named "Data analysis" has been added and the second paragraph of section 3.3 has been moved. Section 3.4 is as follows:*
*"The data are analyzed in a systematic manner. First, the CTR simulation is compared to available observations such as wind speed and direction, temperature, relative humidity, height of the rain-snow transition as well as precipitation amount and types collected during the field project. The time evolution of mass content of ice crystals, cloud, rain, snow pellets and snow are analyzed at the grid point closest to the KES site. To analyse precipitation aloft at KES and across the Kananaskis Valley, a vertical cross section is plotted and the mass content of hydrometeors as well as the vertical air motion are investigated. Second, the CTR simulation is compared to the three sensitivity experiments: the simulation without the temperature change from melting of snow and snow pellets (NO_MLT), the simulation without the temperature change from sublimation (NO_SBL) and the simulation without graupel (NO_GRPL). Finally, the impact of wind direction and precipitation types formed aloft on the precipitation amounts and types reaching the surface is investigated."*

**Comment 5**: Fig. 3: I suggest adding a panel for model reflectivity, corresponding to panel (a). Also, this and all other discussion/figures about the model simulations are from the 1-km domain only, right? This should be stated clearly somewhere.
*A panel showing the simulated radar reflectivity has been added as suggested in Fig. 3 (see specific comment #3). Indeed, figures showing simulated results are for the 1-km domain. This is now stated in section 3.1 as "The following analysis of simulated results will focus only on outputs from the high-resolution domain.".*

**Comment 6**: Fig. 4: According to Fig. 3d, a significant amount of the precipitation was from

snow pellets, with some snow, at least at that location. Perhaps it would be useful to separate the accumulated precip from snow and snow pellets, rather than combining them in Fig. 4a.

*As suggested, accumulated snow and graupel are now presented separately on Fig. 4a and b (see specific comment #4).*

**Comment 7**: The time series plots look quite choppy. Is it possible to output the time series with higher temporal frequency in order to produce smoother plots? (Not a big deal; just a suggestion.)

*The reduced size plots make them look less choppy. Hope that it is satisfactory.*

**Comment 8**: Was there any "hail" in the simulations? Clearly this case does not support hail in nature, but there is a hail category in the MP scheme, which is also used to represent small frozen raindrops. If the model hail mixing ratios are indeed zero in all the simulations, this should be stated and state that for this reason hail is excluded from the figures.

*The referee is right: the simulated hail mixing ratios are zero in all the simulations. A comment was added to section 4.2: "Note that no hail was produced in the simulations."*

**Comment 9**: p. 12, line 6, "Less ice crystals ...[ref to nucleation]. You are not showing number concentration, you are showing mass – there is less ice (crystal category) mass, not fewer number. This is probably not due to nucleation, but rather changes in depositional growth.

*The sentence has been rephrased for clarity as: "Less ice crystal and cloud water mass-mixing ratios are produced aloft compared to CTR. This could be explained by the lack of warming from accretion resulting in colder temperature, which leads to less water vapour depositional growth for ice crystals and cloud droplets and less ice nucleation aloft (e.g. Meyers et al., 1992)."*

**Comment 10**: p. 12, line 15, "...more rain reaches the surface because the environmental temperature is higher..." But it is T_wet, not T, that counts (determines melting), right? In that regard, perhaps it would be good to plot the T_wet = 0C isotherm in Fig. 8. (and Fig. 12)

*Yes, T_wet is associated with melting of ice. We tried plotting it on the revised Fig. 6 (comment #5), which replaced Fig. 8, but it is too close from the surface and, therefore, hard to see. We decided to plot T=0°C instead. It is, however, added to Fig. 12 (see below) since that the schematic does not extend as high vertically.*

[Figure]

**Comment 11**: p. 12, lines 24-26. Suggest omitting paragraph or relocate this as an intro to section 6.

*This paragraph has been moved to the beginning of section 6 and rewritten as: "This section will assess the role of sublimation and snow pellet formation on the vertical and horizontal evolution of precipitation intensity and types in the Kananaskis Valley."*

---

## Author Comment (AC3) · 30 May 2019

**General comments**: This manuscript explores the role of sublimation and riming for a weak precipitation event observed in the Canadian Rockies. The study is done with the WRF model using 1 km horizontal grid spacing and a bulk microphysics scheme. The authors made comparisons to data observed at a single site in order to constrain the model simulation. Then, sensitivity tests were performed in order to quantify the impacts of the melting of snow, the sublimation of solid precipitation and the snow pellet formation on the precipitation features. The main conclusion of this study is that the sublimation can have an important impact on the precipitation evolution in a sub-saturated environment at low elevations.

This manuscript is logically presented and the scientific approach is clear. However, few figures need to be improved (see below). Also, the authors need to add some discussions about the limitations of this study since the comparisons between the model and the observed data are performed at a single site and the conclusions are based on a single case study. Moreover, even if the campaign and the numerical tools are clearly referenced, essential details for this study are missing in the manuscript (see below). The manuscript could be published with some improvements to the presentation and more discussion.

*We thank Referee #3 for his/her suggestions and comments, which helped improving the manuscript. Specific and minor comments are addressed point by point below.*

Specific comments:

**Comment 1**: This study focuses on the roles of the sublimation, melting and riming processes but details about the microphysics parameterizations used are missing. What are the assumptions used to represent the ice species, the conversion between each species, the terminal velocities. . . then all these assumptions need to be considered in the discussion/explanation of the main results.

*More details have been added in section 3.2 and the description of the modifications made on the cloud microphysics scheme are now included. Moreover, it is now clearly stated that our study aims to identify qualitatively physical processes responsible for the types and distribution of the precipitation observed at the surface. In this context, we think that our main results and conclusions are not dependent on the specific cloud microphysics scheme used. This is confirmed by a test we made with the available Thompson et al. (2008) scheme showing similar results. See response to Referee #2, Specific comment #1.*

**Comment 2**: The local heating/cooling rates associated to the sublimation, the melting and the riming processes are proportional to the mass. It is probably most relevant to plot the mass content of the different species instead of the mixing ratio. Also, the heating/cooling rates can probably be useful to the discussion. You can plot, for example, the vertical profiles of the diabatic heating rate due to microphysics for the different sensitivity tests.

*The mass content is now plotted on all the figures that include hydrometeor fields. Since the mass content is related to the diabatic heating/cooling, we believe they were not necessary here. As an example, we included Figs. 15 and 18 from Emilie Poirier's MSc thesis available online. These figures show the time series at KES of the heating/cooling associated with the CTR and*

*NO_GRPL runs.*

[Figure]

[Figure]

**Figure 1.15** Cooling rate ($dT/dt$) associated with (a) melting and (b) sublimation of snow above the KES site on 31 March 2015 for the run assuming no graupel formation. The solid line indicates the height of the 0°C isotherm and the dashed line indicates the height where the wet-bulb temperature is 0°C.

**Figure 1.8** Cooling rate ($dT/dt$) associated with (a) melting and (b) sublimation of snow and graupel above the KES site on 31 March 2015 for the control run. The solid line indicates the height of the 0°C isotherm and the dashed line indicates the height where the wet-bulb temperature is 0°C. The red color indicates an area of heating due to vapor deposition.

*These show that the cooling is on the same order of magnitude for both runs. We also see that the diabatic cooling from sublimation occurred at all temperature whereas diabatic cooling from melting occurred below the 0°C isotherm and is delimited by the 0°C wet bulb temperature isotherm. As mentioned previously, the mass content fields show similar behaviours.*

**Comment 3**: The description of the campaign and the available instruments/observations need to be expanded and clarified. For example, Fig 1 shows different sites but the data used in the manuscript were primarily observed at KES. Are there observations available at the other sites? Also, many relevant details for this study are only available in Thériault et al. (2018) and need to be included in this manuscript. It could be interesting to provide a list of the used instruments, the location, the limitations, the observed parameters and the associated references. For examples, the MMR2 gives the temporal evolution of the vertical profile of the reflectivity and Doppler velocity, and the measurement is affected by the signal attenuation due to e.g. the bright band. Finally, the Parsivel optical disdrometer is mentioned but it is never explained how this instrument is useful. It seems, considering the paper of Thériault et al. (2018), that this instrument is used in order to define the type of the surface precipitation. The different methods (automatic and manual) should be briefly described or at least the authors should specify which one is the most accurate in their opinion.

*The main site was KES but "car-sonde" was performed along Fortress Mountain (FOR) during rain snow transition event. Detailed information about the instruments has been added to the 2 paragraph of section 2: "Most of the observations were collected at the Kananaskis Emergency Services (KES) site located a few kilometers southeast of the Nakiska ski area (NAK) and about 15 km south of the Barrier Lake research station (BAR) (Fig. 1). To characterize the atmospheric*

*conditions (temperature and relative humidity) aloft, sounding system was used and balloons were launched at every 3 h during precipitation events. The precipitation layer aloft was characterized using a Micro Rain Radar 2 (MRR2, Klugmann et al., 1996). It gives the temporal evolution of the vertical profile of the reflectivity and Doppler velocity, and the measurement is affected by the signal attenuation due to e.g. the bright band. Basic meteorological measurements were also available (pressure, wind speed and direction, temperature, dew point temperature). A GEONOR weighing precipitation gauge (Rasmussen et al., 2012) was used to measure the liquid equivalent amount of precipitation. An OTT Parsivel 2 (Battaglia et al., 2010) optical disdrometer was used to characterize the type of hydrometeor because it measures the fallspeed and diameter of precipitation particles. Manual observations of weather conditions including precipitation types were also reported in a systematic manner. In addition, precipitation types are automatically diagnosed using the Ishizaka et al. (2013) method also used in Thériault et al. (2018). The manual method is more precise because one can estimate the degree of riming and the exact crystal types. The Ishizaka et al. (2013) method gives a good idea of the degree of riming but it is not possible to diagnose the type of ice crystal because of the bin sizes. Vertical profiles of basic meteorological features were also obtained using a Kestrel attached to a ski pole and a GPS (Thériault et al., 2014) at two other sites to characterize rain-snow transitions at NAK and at Fortress Mountain (FOR). Further details about the field campaign are given in Thériault et al. (2018). "*

*Klugmann, D., Heinsohn, K., and Kirtzel, H.: A low cost 24 GHz FM-CW Doppler radar rain profiler, Contr. Atmos. Phys., 69, 247–253, 1996.*

**Comment 4**: The parameterizations of the microphysics processes evaluated in this study as well as the modifications made to the bulk microphysics scheme of Milbrandt and Yau (2005a,b) should be described in the section 3.2 of the manuscript.

*This comment was addressed in 2 steps:*

1) *The modifications made to the scheme are described in section 3.2. These sentences have been added: "Given that graupel can sublimate at temperatures >0°C, the same equation was used for snow, which is*

$$QVD_{vs} = \frac{1}{AB_i} \left[ 2\pi(S_i - 1)N_{0s}VENT_s - \frac{L_s L_f}{K_a R_v T^2}QCL_{cs} \right]$$

*where*

$$AB_i = \frac{L_s^2}{K_a R_v T^2} + \frac{1}{\rho q_{is}\psi}$$

*is the thermodynamic function. Also, $S_i$ is the saturation ratio with respect to ice, $N_{0S}$ is the intercept parameter for snow, $VENT_S$ is the mass-weighted ventilation factor (Ferrier, 1994), $K_a$ is the thermal conductivity of air, $R_v$ is the gas constant for water vapor, $T$ is the temperature of air, $\rho$ is the density of air, $q_{is}$ is the saturation vapor mixing ratio with respect to ice and $\psi$ is the diffusivity of water vapor in air.*

*The sublimation rate equation was moved in the microphysics scheme so that snow and graupel sublimation are computed in the same conditions, which is at all air temperatures. The function polysvp was also corrected in the microphysics scheme to calculate the saturation vapor pressure properly at all temperatures. This bug was fixed in the following version of WRF"*

2) *A description of the simulations conducted and the microphysical processes studies are described in the new section 3.3 called Description of the sensitivity experiment. This is the new section:*

   *"The control simulation (CTR) is conducted using the modified microphysics and model configuration described in section 3.1 and 3.2. To estimate the impact of temperature changes while neglecting the diabatic heating/cooling due to the precipitation phase transition and no graupel formation, the three following sensitivity experiments were performed:*

   a. *The diabatic cooling of melting snow and graupel was neglected. Hence, snow and graupel were allowed to melt into rain but no energy was extracted from the environment to melt the particles. This experiment is called NO_MLT.*

   b. *The diabatic cooling of sublimation was neglected. In a similar manner as for MLT, snow and graupel were allowed to sublimate but the temperature interaction with the environment was not taken into account. This experiment is called NO_SBL.*

   c. *Since that graupel was often reaching the surface at KES during the Alberta field project (Thériault et al., 2018), another simulation was performed. The initiation of graupel was suppressed to turn the production of graupel off. It was also ensured that there were no sources or sinks, hence, no warming from the cloud droplets freezing on the solid particles (snow or/and ice) and no sublimation of graupel since none was produced. This experiment is called NO_GRPL."*

**Comment 5**: The comparison between the CTL simulation and the observations should be discussed in more details, especially the vertical structure. The vertical profile of temperature and dew point temperature obtained in CTL is plotted in Figure 2 but never mentioned in the manuscript.

*A discussion was added to the 2ⁿᵈ paragraph of section 4.1. "Concerning the general meteorological parameters, the CTR run show similar patterns than the observations at KES (cf. Figs. 2 and 3). The vertical structure of the temperature and dewpoint are similar but the model is mainly colder and moister than the observations. The wet-bulb temperature is, however, similar (Fig 2.)". The figure was added to minor comment #7.*

The temporal evolution of the vertical profile of the precipitation field observed at KES is given in Fig3a but not compared with the simulation results; at least qualitatively due to the signal attenuation due to the bright band (Matrosov, 2008). MRR2 also provides the Doppler velocity fields; is it possible to compare and assess the species fall speed simulated in the CTL run?

*The simulated reflectivity field was added to Fig. 3 and a comparison is done in the text between reflectivity fields from the MMR2 and the simulated one in section 4.1, which has been completely rewritten, see response to Referee 2 specific comment #3. However, we did not think that it was necessary to plot the Doppler velocity, but it would be interesting if the goal of the study would be to do a model comparison with observations.*
*This is the revised Fig. 3:*

[Figure]

*Figure 3 (revised): Atmospheric conditions and precipitation fields during the 31 March 2015 event at KES. (a) Reflectivity field measured by the Micro Rain Radar and (b) is estimated by the model (CTR). Reflectivity values > 30dBZ are associated with the radar reflectivity bright-band.; (c) surface temperature (T) and relative humidity (RH) observed (black line) and simulated (blue line); (d) wind speed and direction using wind barbs, where the observed is black and simulated is blue. An empty circle is wind speed rounded at 0 knots, a short bar is rounded at 5 knots; (e) unadjusted liquid equivalent accumulated precipitation observed (black line, OBS) and simulated (bold blue line for total, green line for rain, thin blue line for graupel and dashed blue line for snow), and (f) the type of precipitation observed manual (MAN) and automatically (AUT) at KES.*

*These are rain (R), graupel (GR), snow (S), mixed precipitation (M), heavily rimed snow (HR), rimed aggregates (RA), dry snow (DS) and dendrites (DE). Simulated results are for the CTL run. Adapted from Thériault et al. (2018).*

Moreover, it is stated several times that the model well reproduces the surface observations. You should say that the CTL simulations reasonably reproduce the observations in order to perform sensitivity studies. However, few parameters simulated in CTL differ from the observations. Indeed, a time shift is visible in the temporal evolution of the accumulated precipitation and the temperature.

*It is now explained that we want to demonstrate that simulated results compare well with observations in a qualitative point of view and not quantitative. First sentence in section 4 was modified as: "The CTR simulation is compared to observations to ensure that atmospheric conditions are sufficiently well represented by the model to ensure its use as a qualitative analysis tool of physical processes.". The time shift in the temporal evolution of the accumulated precipitation and temperature is now commented in section 4.1.*

Do you estimate the impact on the results of this comparison between observations and CTL simulations if you choose another grid box?
*We did try other grid point but the one used in this study compared better with observations.*

**Comment 6**: The figures used to illustrate the sensitivity tests are difficult to interpret. I suggest plotting the differences between the CTL simulations and each sensitivity results.
*We prefer showing the full mass content fields instead of differences and new figures have been done for clarity (Figs. 5 and 6 as well as 11, whereas Figs. 7, 8, 9 are deleted) Hope that you will find our new figures easier to interpret.*

**Comment 7**: There are spelling and grammar errors throughout the manuscript. I suggest that the authors read through it carefully and clean it up before resubmitting.
*The manuscript was carefully reread to check for language issues.*

Minor comments :
**Comment 1**: P1-L24. "rain-snow boundary" term is used but defined in the following paragraph. Moreover, the definition is confusing. It is equivalent to the radar bright band/melting layer?
*The "rain-snow boundary" in that sentence has been changed to "0°C isotherm" for clarity. Also, the definition of rain-snow transition was improved as follows: "The top of the boundary corresponds to the top of the melting layer aloft, which is represented by the radar bright band (Fabry and Zawadzki, 1995) and the base of the boundary is when all solid precipitation has melted into rain."*

**Comment 2**: P2-L8. Sometimes the term solid precipitation is used. Is it ice-phase precipitation? Or precipitation with high density≠ snow?

*Solid precipitation was changed for ice-phase precipitation throughout the text.*

**Comment 3**: P2-L12. "using numerical simulations" What type of simulations: 1D, 2D, 3D?

*It is a 3D simulation, this is now indicated in the text.*

**Comment 4**: P2-L23/28. These studies were performed over mountainous area?

*This paragraph is to highlight the studies of precipitation in relatively dry conditions. The first sentence of that paragraph was changed to: "Few studies have examined precipitation features in northern Canada, in relatively dry areas.".*

**Comment 5**: P3-Fig1. The title of the right panel is not clear. Do you mean domain area?

*Titles above panels were deleted (see the revised figure in reply of comment #18).*

**Comment 6**: P3-L11. The name of the field campaign is only given in the conclusions section and should be mentioned here.

*The first sentence now reads: "During the Alberta Field Project held in the Kananaskis Valley in March-April 2015, …".*

**Comment 7**: P4-Fig2. The authors may consider adding details on the skewT-logP diagram in order to define the Lifted Condensation Level and Tw.

*Figure 2 has now been completed. This is the revised Fig. 2. The blue tone lines are the CTR and the black/grey tones are the observations.*

[Figure]

*Figure 2. Vertical profiles of air temperature (T, solid line), dew point temperature (Td, dashed line) and wet-bulb temperature (Tw, light colour) at 2100 UTC 31 March 2015 at the KES site. The measurement (OBS) and the control simulation (model) described in section 3.1, are represented by blue and black/grey lines, respectively.*

**Comment 8**: P4- L9. The relative humidity is never given in the manuscript. If available, the temporal evolution of the relative humidity should be added to the Fig 3.

*The relative humidity of measured and simulated has been added to Figure 3 instead of the dew point temperature (see response to specific comment #5).*

**Comment 9**: P4-L10. The bright-band is close to the surface at the KES station?

*The sentence was clarified: "The bright-band is located at the elevation where precipitation started to melt."*

**Comment 10**: P5-L1. "brief period of only snow". According to Fig3e, there is no S period?

*Snow has been changed to rain in the text.*

**Comment 11**: Fig3. You should increase the y- axis because the reader may have difficulties to extract the values, for example for the temperature. Few elements are missing in the caption: wind barbs definition, hatched region in fig 3a.

*Figure 3 has been improved and the legend has been completed. Also, simulated reflectivity has been added (see response to specific comment #5).*

**Comment 12**: P6-L5. The boundary conditions forcing is every 3h, 6h or 12h?

*This is now indicated as: "The boundary conditions forcing is done every 3 hours.".*

**Comment 13**: P6-L7. Add the number of grid points of the innermost domain in order to have an idea of the surface area.

*The number of grid points of the innermost domain has been added in section 3.1: "The high-resolution domain is shown in Fig. 1; it has 118 x 106 grid points.".*

**Comment 14**: P6-L23. The most common term is probably "graupel" instead of "snow pellet".

*It has been changed throughout the manuscript.*

**Comment 15**: P7-L3/18. The section 3.3 can be summarized in one paragraph because the setup of the sensitivity tests is given twice. Also, the first sentence of the section 4 explains that the CTL simulations will be compared to the available observations described in the previous section.

*We chose to keep section 3.3 but to change the beginning of section 5 as: "The roles of phase changes and of the production of graupel on precipitation amounts and types reaching the surface at KES are investigated by comparing the CTR simulation with sensitivity experiments (NO_MLT; NO_SUB; NO_GRPL).". For section 4, it now begins as: "The CTR simulation is compared to observations to ensure that atmospheric conditions are sufficiently well represented by the model to ensure its use as a qualitative analysis tool of physical processes."*

**Comment 16**: P8-L5. . . .. also investigated "in the CTL simulations"
*Section 4 has been separated in two sub-sections for clarity.*

**Comment 17**: P8-L10. Fig3e indicates a much shorter period of snow pellet precipitation
*The manual observations reported some mixed precipitation, which could include graupel and/or snow mixed with rain. This is now better described in section 2 as "Manual observations at the KES site show that light rain started at 2030 UTC 31 March 2015, changing to a mixture of rain, snow and graupel between about 2150 UTC and 2215 UTC, then to a brief period of only rain (Fig. 3e).*

**Comment 18**: P8-L11. Indicate on Fig 1. or on Fig 4 where is the cross section plotted on Figures 6, 9 and 11.
*Figure 1 has been modified to indicate the position of the cross section (red line) plotted on Figs. 6 and 11 (Fig. 9 has been deleted).*

[Figure]

*Figure 1. Area of interest (left) and 1km mesh domain (right) used for the numerical simulations with the WRF model. BAR stands for the Barrier Lake research station, NAK for Nakiska ski area, KES for the Kananaskis Emergency Services site and FOR for Fortress Mountain. Red line on the right panel indicates the position of the cross section used in Figs. 6, 9 and 11.*

**Comment 19**: P8-L15/16. Why the vertical movements would initiate only ice crystals and cloud droplets and not the other species? The amount of snow and snow pellet seem much larger than the amount of ice crystals and cloud and rain water? What is the role of deposition?
*The vertical movement over the western barrier corresponds to air ascent, which favours heterogeneous nucleation of ice crystals and cloud droplets. It is the combination of ice crystals and cloud droplets, which initiate the snow formation. The water vapour depositional growth plays a role as this growth favours the formation of snow, which is more efficient for larger particles (either droplets or ice crystals). Clarifying Figure 6 is now helpful to answer these points (see response to comment #20).*

**Comment 20**: Fig.6. The intensity of the vertical wind is difficult to read. Also, the definition of the dashed/solid lines for wind is missing in the caption. The wet bulb temperature is plotted but not mentioned in the Section

*Figure 6 has been redone and the definition of dashed/solid lines for wind has been added in the caption. The temperature was plotted and not the wet-bulb temperature. It has been corrected in the new caption. This is the revised Fig. 6.*

[Figure]

*Figure 6: Comparison of the vertical cross-section across the Kananaskis Valley along the red line in Fig. 1 showing the mass content of hydrometeors during the 4 simulations conducted for CTR, NO_MLT, NO_SBL and NO_GRPL from left to right. (a-d) is ice mass content (x10 g/kg) with vertical velocity (m/s). The yellow line is 0 m/s, the dashed lines are negative values and solid lines are positive values, (e-h) is clouds and rain mass content, (i-l) is graupel mass content and (m-p) is snow mass content. The 0°C isotherm is indicated by the solid black line. Panels a-p have the same colour scale. The location of KES is indicated by the purple dot.*

**Comment 21**: P11-L14. Why do you choose this threshold in order to plot the Fig8? Fig8 is difficult to interpret; you should make a difference between CTL and each sensitivity test.

*Figure 8 was deleted. The information is given in Fig. 5. This is the revised Fig. 5.*

[Figure]

*Figure 5 (revised): Comparison of the time evolution of hydrometeors at the surface and aloft at KES during the 4 simulations conducted for CTR, NO_MLT, NO_SBL and NO_GRPL from left to right. (a-d) is ice mass content (x10 g/kg), (e-h) is clouds and rain mass content, where rain is only formed through melting of ice, so it is only present near the surface, (i-l) is graupel mass content, (m-p) is snow mass content and (q-t) is the surface precipitation rate of rain (R), graupel (G) and snow (S). The 0°C isotherm is indicated by the solid black line on (a-p). Panels a-p have the same colour scale.*

**Comment 22**: P12-L8. I do not understand the nucleation citation. You never work with the concentration parameter?

*The heterogeneous ice nucleation is parameterized using Meyers et al. (1992), which gives a number of pristine ice crystals formed depending on air temperature and ice supersaturation. Then, an assumed size of newly formed ice crystals allows computing the mass content of pristine ice formed.*

**Comment 23**: P13-L8. "flow reversal". Do you mean wind shear?

*Yes, the term "flow reversal" was replaced by "wind shear" for clarity.*